# Gaussian Mixture Flow Matching Models

**Hansheng Chen** [1]   **Kai Zhang** [2]   **Hao Tan** [2]   **Zexiang Xu** [3]   **Fujun Luan** [2]
**Leonidas Guibas** [1]   **Gordon Wetzstein** [1]   **Sai Bi** [2]

https://github.com/Lakonik/GMFlow

## Abstract

Diffusion models approximate the denoising distribution as a Gaussian and predict its mean, whereas flow matching models reparameterize the Gaussian mean as flow velocity. However, they underperform in few-step sampling due to discretization error and tend to produce over-saturated colors under classifier-free guidance (CFG). To address these limitations, we propose a novel Gaussian mixture flow matching (GM-Flow) model: instead of predicting the mean, GM-Flow predicts dynamic Gaussian mixture (GM) parameters to capture a multi-modal flow velocity distribution, which can be learned with a KL divergence loss. We demonstrate that GMFlow generalizes previous diffusion and flow matching models where a single Gaussian is learned with an $L_2$ denoising loss. For inference, we derive GM-SDE/ODE solvers that leverage analytic denoising distributions and velocity fields for precise few-step sampling. Furthermore, we introduce a novel probabilistic guidance scheme that mitigates the over-saturation issues of CFG and improves image generation quality. Extensive experiments demonstrate that GMFlow consistently outperforms flow matching baselines in generation quality, achieving a Precision of 0.942 with only 6 sampling steps on ImageNet 256×256.

## 1. Introduction

Diffusion probabilistic models (Sohl-Dickstein et al., 2015; Ho et al., 2020), score-based models (Song & Ermon, 2019; Song et al., 2021b), and flow matching models (Lipman et al., 2023; Liu et al., 2022) form a family of generative

[1]Stanford University, CA 94305, USA [2]Adobe Research, CA 95110, USA [3]Hillbot. Correspondence to: Hansheng Chen <hanshengchen@stanford.edu>.

*Proceedings of the $42^{nd}$ International Conference on Machine Learning*, Vancouver, Canada. PMLR 267, 2025. Copyright 2025 by the author(s).

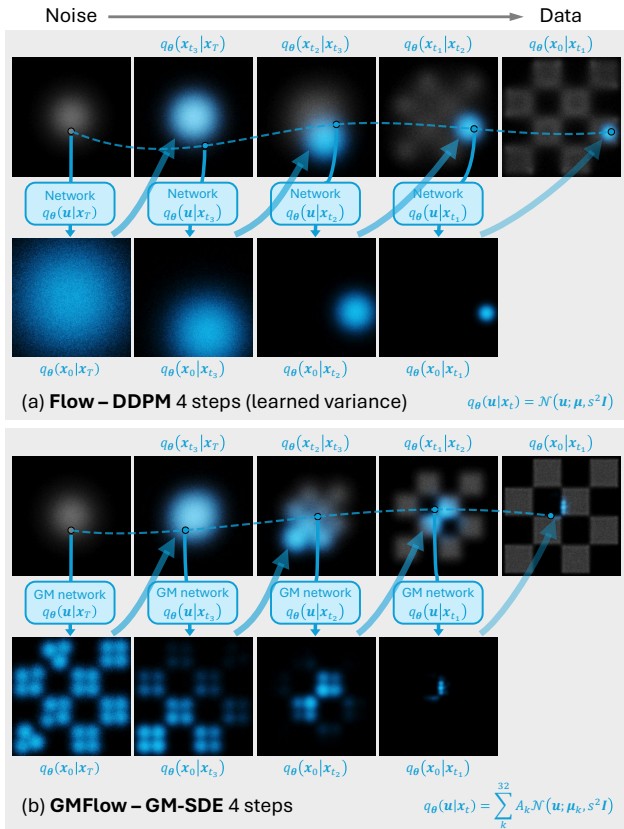

*Figure 1.* Comparison between vanilla diffusion (with flow velocity parameterization) and GMFlow on a 2D checkerboard distribution. (a) The vanilla diffusion model predicts mean velocity, modeling the denoising distribution $q_\theta(\boldsymbol{x}_0|\boldsymbol{x}_t)$ as a single Gaussian, and then samples from a Gaussian transition distribution $q_\theta(\boldsymbol{x}_{t-\Delta t}|\boldsymbol{x}_t)$. (b) GMFlow predicts a GM for velocity and yields a multi-modal GM denoising distribution, from which the transition distribution can be analytically derived for precise next-step sampling, allowing more accurate few-step sampling (4 steps in this case).

models that share an underlying theoretical framework and have made significant advances in image and video generation (Yang et al., 2023; Po et al., 2024; Rombach et al., 2022; Saharia et al., 2022b; Podell et al., 2024; Chen et al., 2024; Esser et al., 2024; Blattmann et al., 2023; Hong et al., 2023; HaCohen et al., 2024; Kong et al., 2025). Standard

diffusion models approximate the denoising distribution as a Gaussian and train neural networks to predict its mean, and optionally its variance (Nichol & Dhariwal, 2021; Bao et al., 2022b;a). Flow matching models reparameterize the Gaussian mean as flow velocity, formulating an ordinary differential equation (ODE) that maps noise to data. These formulations remain dominant in image generation as per user studies (Artificial Analysis, 2025; Jiang et al., 2024).

However, vanilla diffusion and flow models require tens of sampling steps for high-quality generation, since Gaussian approximations hold only for small step sizes, and numeric ODE integration introduces discretization errors. In addition, high quality generation requires a higher classifier-free guidance (CFG) scale (Ho & Salimans, 2021), yet stronger CFG often leads to over-saturated colors (Saharia et al., 2022a; Kynkäänniemi et al., 2024) due to out-of-distribution (OOD) extrapolation (Bradley & Nakkiran, 2024), thus limiting overall image quality even with hundreds of steps.

To address these limitations, we deviate from previous single-Gaussian assumption and introduce Gaussian mixture flow matching (GMFlow). Unlike vanilla flow models, which predict the mean of flow velocity $\boldsymbol{u}$, GMFlow predicts the parameters of a Gaussian mixture (GM) distribution, representing the probability density function (PDF) of $\boldsymbol{u}$. This provides two key benefits: (a) our GM formulation captures more intricate denoising distributions, enabling more accurate transition estimates at larger step sizes and thus requiring fewer steps for high-quality generation (Fig. 1); (b) CFG can be reformulated by reweighting the GM probabilities rather than extrapolation, thus bounding the samples within the conditional distribution and avoiding over-saturation, thereby improving overall image quality.

We train GMFlow by minimizing the KL divergence between the predicted velocity distribution and the ground truth distribution, which we show is a generalization of previous diffusion and flow matching models (§ 3.1). For inference, we introduce novel SDE and ODE solvers that analytically derive the reverse transition distribution and flow velocity field from the predicted GM, enabling fast and precise few-step sampling (§ 3.3). Meanwhile, we develop a probabilistic guidance approach for conditional generation, which reweights the GM PDF using a Gaussian mask to enhance condition alignment (§ 3.2).

For evaluation, we compare GMFlow against vanilla flow matching baselines on both 2D toy dataset and ImageNet (Deng et al., 2009). Extensive experiments reveal that GM-Flow consistently outperforms baselines equipped with advanced solvers (Lu et al., 2022; 2023; Zhao et al., 2023; Karras et al., 2022). Notably, on ImageNet 256×256, GM-Flow excels in both Precision and FID metrics with fewer than 8 sampling steps; with 32 sampling steps, GMFlow achieves a state-of-the-art Precision of 0.950 (Fig. 6).

The main contributions of this paper are as follows:

- We propose GMFlow, a generalized formulation of diffusion models based on GM denoising distributions.
- We introduce a GM-based sampling framework consisting of novel SDE/ODE solvers and probabilistic guidance.
- We empirically validate that GMFlow outperforms flow matching baselines in both few- and many-step settings.

## 2. Diffusion and Flow Matching Models

In this section, we provide background on diffusion and flow matching models as the basis for GMFlow. Note that we introduce flow matching as a special diffusion parameterization since they largely overlap in practice (Albergo & Vanden-Eijnden, 2023; Gao et al., 2024).

**Forward diffusion process.** Let $\boldsymbol{x} \in \mathbb{R}^D$ be a data point sampled from a distribution with its PDF denoted by $p(\boldsymbol{x})$. A typical diffusion model defines a time-dependent interpolation between the data point and a random Gaussian noise $\boldsymbol{\epsilon} \sim \mathcal{N}(\boldsymbol{0}, \boldsymbol{I})$, yielding the noisy data $\boldsymbol{x}_t = \alpha_t \boldsymbol{x} + \sigma_t \boldsymbol{\epsilon}$, where $t \in [0, T]$ denotes the diffusion time, and $\alpha_t, \sigma_t$ are the predefined time-dependent monotonic coefficients (noise schedule) that satisfy the boundary condition $\boldsymbol{x}_0 = \boldsymbol{x}$, $\boldsymbol{x}_T \approx \boldsymbol{\epsilon}$. Apparently, the marginal PDF of the noisy data $p(\boldsymbol{x}_t)$ can be written as the data distribution $p(\boldsymbol{x})$ convolved by a Gaussian kernel $p(\boldsymbol{x}_t | \boldsymbol{x}_0) = \mathcal{N}(\boldsymbol{x}_t; \alpha_t \boldsymbol{x}_0, \sigma_t^2 \boldsymbol{I})$, i.e., $p(\boldsymbol{x}_t) = \int_{\mathbb{R}^D} p(\boldsymbol{x}_t | \boldsymbol{x}_0) p(\boldsymbol{x}_0) \, \mathrm{d}\boldsymbol{x}_0$. Alternatively, $p(\boldsymbol{x}_t)$ can be expressed as a previous PDF $p(\boldsymbol{x}_{t-\Delta t})$ convolved by a transition Gaussian kernel $p(\boldsymbol{x}_t | \boldsymbol{x}_{t-\Delta t})$, given by:

$$p(\boldsymbol{x}_t | \boldsymbol{x}_{t-\Delta t}) = \mathcal{N}\left( \boldsymbol{x}_t; \frac{\alpha_t}{\alpha_{t-\Delta t}} \boldsymbol{x}_{t-\Delta t}, \overbrace{\left( \sigma_t^2 - \frac{\alpha_t^2}{\alpha_{t-\Delta t}^2} \sigma_{t-\Delta t}^2 \right)}^{\beta_{t,\Delta t}} \boldsymbol{I} \right), \quad (1)$$

where the transition variance is denoted by $\beta_{t,\Delta t}$ for brevity. A series of convolutions from $0$ to $T$ constructs a diffusion process over time. With infinitesimal step size $\Delta t$, this process can be continuously modeled by the forward-time SDE (Song et al., 2021b).

**Reverse denoising process.** Diffusion models generate samples by first sampling the noise $\boldsymbol{x}_T \leftarrow \boldsymbol{\epsilon}$ and then reversing the diffusion process to obtain $\boldsymbol{x}_0$. This can be achieved by recursively sampling from the reverse transition distribution $p(\boldsymbol{x}_{t-\Delta t} | \boldsymbol{x}_t) = \frac{p(\boldsymbol{x}_{t-\Delta t}) p(\boldsymbol{x}_t | \boldsymbol{x}_{t-\Delta t})}{p(\boldsymbol{x}_t)}$. DDPM (Ho et al., 2020) approximate $p(\boldsymbol{x}_{t-\Delta t} | \boldsymbol{x}_t)$ with a Gaussian and reparameterize its mean in terms of residual noise, which is then learned by a neural network. Its sampling process is equivalent to a first-order solver of a reverse-time SDE. Song et al. (2021b) reveal that the time evolution of the marginal distribution $p(\boldsymbol{x}_t)$ described by the SDE is the same as that described by a flow ODE, which maps the noise $\boldsymbol{x}_T$ to data $\boldsymbol{x}_0$ deterministically. In particular, flow matching models

train a neural network to directly predict the ODE velocity field $\frac{d\boldsymbol{x}_t}{dt}$. They typically adopt a linear noise schedule, which defines $T := 1, \alpha_t := 1 - t, \sigma_t := t$, thus yielding a simplified velocity formulation $\frac{d\boldsymbol{x}_t}{dt} = \mathbb{E}_{\boldsymbol{x}_0 \sim p(\boldsymbol{x}_0|\boldsymbol{x}_t)}[\boldsymbol{u}]$, with the random flow velocity $\boldsymbol{u}$ defined as:

$$\boldsymbol{u} := \frac{\boldsymbol{x}_t - \boldsymbol{x}_0}{\sigma_t}. \tag{2}$$

This reveals that the velocity field $\frac{d\boldsymbol{x}_t}{dt}$ represents the mean of the random velocity $\boldsymbol{u}$ over the denoising distribution $p(\boldsymbol{x}_0|\boldsymbol{x}_t) = \frac{p(\boldsymbol{x}_0)p(\boldsymbol{x}_t|\boldsymbol{x}_0)}{p(\boldsymbol{x}_t)}$.

**Flow matching loss.** Flow models stochastically regress $\frac{d\boldsymbol{x}_t}{dt}$ using randomly paired samples of $\boldsymbol{x}_0$ and $\boldsymbol{x}_t$. Let $\boldsymbol{\mu_\theta}(\boldsymbol{x}_t)$ denote a flow velocity neural network with learnable parameters $\boldsymbol{\theta}$, the $L_2$ flow matching loss is given by:

$$\mathcal{L} = \mathbb{E}_{t,\boldsymbol{x}_0,\boldsymbol{x}_t}\left[\frac{1}{2}\|\boldsymbol{u} - \boldsymbol{\mu_\theta}(\boldsymbol{x}_t)\|^2\right]. \tag{3}$$

In practice, additional condition signals $\boldsymbol{c}$ (e.g., class label or text prompt) can be fed to the network $\boldsymbol{\mu_\theta}(\boldsymbol{x}_t, \boldsymbol{c})$, making it a conditioned diffusion model that learns $p(\boldsymbol{x}_0|\boldsymbol{c})$.

**Limitations.** Sampling errors in flow models can arise from two sources: (a) discretization errors in SDE/ODE solvers, and (b) inaccurate flow prediction $\boldsymbol{\mu_\theta}(\boldsymbol{x}_t, \boldsymbol{c})$ due to underfitting (Karras et al., 2024). While discretization errors can be reduced by reducing step size, it increases the number of function evaluations (NFE), causing significant computational overhead. To mitigate prediction inaccuracies, CFG (Ho & Salimans, 2021) performs an extrapolation of conditional and unconditional predictions, given by $w\boldsymbol{\mu_\theta}(\boldsymbol{x}_t, \boldsymbol{c}) + (1 - w)\boldsymbol{\mu_\theta}(\boldsymbol{x}_t)$, where $w \in [1, +\infty)$ is the guidance scale. Such a method improves image quality and condition alignment at the expense of diversity. However, higher $w$ may lead to OOD samples and cause image oversaturation, which is often mitigated with heuristics such as thresholding (Saharia et al., 2022a; Sadat et al., 2025).

## 3. Gaussian Mixture Flow Matching Models

In this section, we introduce our GMFlow models, covering its parameterization and loss function (§ 3.1), probabilistic guidance mechanism (§ 3.2), GM-SDE/ODE solvers (§ 3.3), and other practical designs for image generation (§ 3.4). Table 1 summarizes the key differences between GMFlow and vanilla flow models. Algorithms 1 and 2 present the outlines of training and sampling schemes, respectively.

### 3.1. Parameterization and Loss Function

Different from vanilla flow models that regress mean velocity, we model the velocity distribution $p(\boldsymbol{u}|\boldsymbol{x}_t)$ as a Gaussian

*Table 1.* Comparison between vanilla flow models and GMFlow.

| | **Vanilla diffusion flow** | **GMFlow** |
|---|---|---|
| Transition assumption | $\Delta t$ is small $\to p(\boldsymbol{x}_{t-\Delta t}|\boldsymbol{x}_t)$ is approximately a Gaussian | Derive $p(\boldsymbol{x}_{t-\Delta t}|\boldsymbol{x}_t)$ from the predicted GM |
| Network output | Mean of flow velocity $\boldsymbol{\mu_\theta}(\boldsymbol{x}_t)$ | GM params in $q_\theta(\boldsymbol{u}|\boldsymbol{x}_t)$ $= \sum_{k=1}^K A_k \mathcal{N}(\boldsymbol{u}; \boldsymbol{\mu}_k, s^2\boldsymbol{I})$ |
| Training loss | $\mathbb{E}_{t,\boldsymbol{x}_0,\boldsymbol{x}_t}\left[\frac{1}{2}\|\boldsymbol{u} - \boldsymbol{\mu_\theta}(\boldsymbol{x}_t)\|^2\right]$ | $\mathbb{E}_{t,\boldsymbol{x}_0,\boldsymbol{x}_t}[-\log q_\theta(\boldsymbol{u}|\boldsymbol{x}_t)]$ |
| Sampling methods | 1st-ord: Euler, DDPM... 2nd-ord: DPM++, UniPC... | 1st-ord: GM-SDE/ODE 2nd-ord: GM-SDE/ODE 2 |
| Guidance | Mean extrapolation $w\boldsymbol{\mu_\theta}(\boldsymbol{x}_t, \boldsymbol{c}) + (1 - w)\boldsymbol{\mu_\theta}(\boldsymbol{x}_t)$ | GM reweighting $\frac{w(\boldsymbol{u})}{Z}q_\theta(\boldsymbol{u}|\boldsymbol{x}_t, \boldsymbol{c})$ |

mixture (GM):

$$q_\theta(\boldsymbol{u}|\boldsymbol{x}_t) = \sum_{k=1}^K A_k \mathcal{N}(\boldsymbol{u}; \boldsymbol{\mu}_k, \boldsymbol{\Sigma}_k), \tag{4}$$

where $\{A_k, \boldsymbol{\mu}_k, \boldsymbol{\Sigma}_k\}$ are dynamic GM parameters predicted by a network with parameters $\boldsymbol{\theta}$, and $K$ is a hyperparameter specifying the number of mixture components. To enforce $\sum_k A_k = 1$ with $A_k \geq 0$, we employ a softmax activation $A_k = \frac{\exp a_k}{\sum_k \exp a_k}$, where $\{a_k \in \mathbb{R}\}$ are pre-activation logits.

We train GMFlow by matching the predicted distribution $q_\theta(\boldsymbol{u}|\boldsymbol{x}_t)$ with the ground truth velocity distribution $p(\boldsymbol{u}|\boldsymbol{x}_t)$. Specifically, we minimize the Kullback–Leibler (KL) divergence (Kullback & Leibler, 1951) $\mathbb{E}_{\boldsymbol{u}\sim p(\boldsymbol{u}|\boldsymbol{x}_t)}[\log p(\boldsymbol{u}|\boldsymbol{x}_t) - \log q_\theta(\boldsymbol{u}|\boldsymbol{x}_t)]$, where $\log p(\boldsymbol{u}|\boldsymbol{x}_t)$ does not affect backpropagation and can be omitted. During training, we sample $\boldsymbol{u} \sim p(\boldsymbol{u}|\boldsymbol{x}_t)$ by drawing a pair of $\boldsymbol{x}_0, \boldsymbol{x}_t$, and then calculate the velocity using Eq. (2). The resulting loss function is therefore reformulated as:

$$\mathcal{L} = \mathbb{E}_{t,\boldsymbol{x}_0,\boldsymbol{x}_t}[-\log q_\theta(\boldsymbol{u}|\boldsymbol{x}_t)]. \tag{5}$$

In Eq. (6), we present an expanded form of this loss function.

**Why choosing Gaussian mixture?** While there are a large family of parameterized distributions, we choose Gaussian mixture for its following desired properties:

- **Mean alignment.** The mean of the ground truth distribution $p(\boldsymbol{u}|\boldsymbol{x}_t)$ is crucial since it decides the velocity term in the flow ODE and the drift term in the reverse-time SDE (§ C.1). When the loss in Eq. (5) is minimized (assuming sufficient network capacity), the GM distribution $q_\theta(\boldsymbol{u}|\boldsymbol{x}_t)$ guarantees that its mean aligns with that of $p(\boldsymbol{u}|\boldsymbol{x}_t)$.

**Theorem 3.1.** Given any distribution $p(\boldsymbol{u})$ and any symmetric positive definite matrices $\{\boldsymbol{\Sigma}_k\}$, if $\{a_k^*, \boldsymbol{\mu}_k^*\}$ are the optimal GM parameters w.r.t. with the objective $\min_{\{a_k, \boldsymbol{\mu}_k\}} \mathbb{E}_{\boldsymbol{u}\sim p(\boldsymbol{u})}[-\log \sum_k A_k \mathcal{N}(\boldsymbol{u}; \boldsymbol{\mu}_k, \boldsymbol{\Sigma}_k)]$, then the GM mean satisfies $\sum_k A_k^* \boldsymbol{\mu}_k^* = \mathbb{E}_{\boldsymbol{u}\sim p(\boldsymbol{u})}[\boldsymbol{u}]$.

We provide the proof of the theorem in § C.2. It's worth pointing out that not all distributions satisfy this property

(e.g. the mean of Laplace distribution aligns to the median instead of the mean).

- **Analytic calculation.** GM enables necessary calculations to be approached analytically (e.g., for deriving the mean and the transition distribution), as detailed in § D.

- **Expressiveness.** GMs can approximate intricate multimodal distributions. With sufficient number of components, GMs are theoretically universal approximators (Huix et al., 2024).

**Simplified covariances.** The expansion of Eq. (5) involves the inverse covariance matrix $\boldsymbol{\Sigma}_k^{-1}$, which can lead to training instability. To mitigate this, we simplify each covariance to a scaled identity matrix, i.e., $\boldsymbol{\Sigma}_k = s^2\boldsymbol{I}$, where $s \in \mathbb{R}_+$ is the predicted standard deviation shared by all mixture components. It's worth pointing out that this simplification does not limit the GM's expressiveness, which is mainly dominated by $K$. Moreover, Theorem 3.1 implies that the structure of the covariance matrix is irrelevant to the accuracy of the mean velocity, mitigating the need for intricate covariances. Under this simplification, the expanded GM KL loss is reduced to:

$$\mathcal{L} = \mathbb{E}_{t,\boldsymbol{x}_0,\boldsymbol{x}_t}\Bigg[ -\log\sum_{k=1}^K \exp\Bigg($$
$$-\frac{1}{2s^2}\|\boldsymbol{u} - \boldsymbol{\mu}_k\|^2 - D\log s + \log A_k\Bigg)\Bigg], \quad (6)$$

which can be interpreted as a hybrid of regression loss to the centroids and classification loss to the components.

**Special cases of GMFlow.** GMFlow generalizes several formulations of previous diffusion and flow models. In a special case where $K = 1, s = 1$, Eq. (6) is identical to the $L_2$ loss in Eq. (3). Therefore, GMFlow is a generalization of vanilla diffusion and flow models. In another case where $\{\boldsymbol{\mu}_k\}$ are velocities towards predefined tokens and $s \approx 0$, then $\{A_k\}$ represent token probabilities, making Eq. (6) analogous to categorical diffusion objectives (Gu et al., 2022; Dieleman et al., 2022; Campbell et al., 2022).

**$\boldsymbol{u}$-to-$\boldsymbol{x}_0$ reparameterization.** While the neural network directly outputs the $\boldsymbol{u}$ distribution, we can flexibly reparameterize it into an $\boldsymbol{x}_0$ distribution by substituting $\boldsymbol{u} = \frac{\boldsymbol{x}_t - \boldsymbol{x}_0}{\sigma_t}$ into Eq. (4), yielding $q_{\boldsymbol{\theta}}(\boldsymbol{x}_0|\boldsymbol{x}_t) = \sum_k A_k\mathcal{N}(\boldsymbol{x}_0; \boldsymbol{\mu}_{\mathrm{x}k}, s_{\mathrm{x}}^2\boldsymbol{I})$, with the new parameters $\boldsymbol{\mu}_{\mathrm{x}k} = \boldsymbol{x}_t - \sigma_t\boldsymbol{\mu}_k$ and $s_{\mathrm{x}} = \sigma_t s$. The velocity KL loss is thus equivalent to an $\boldsymbol{x}_0$ likelihood loss $\mathcal{L} = \mathbb{E}_{t,\boldsymbol{x}_0,\boldsymbol{x}_t}[-\log q_{\boldsymbol{\theta}}(\boldsymbol{x}_0|\boldsymbol{x}_t)]$.

### 3.2. Probabilistic Guidance via GM Reweighting

Vanilla CFG suffers from over-saturation due to unbounded extrapolation, which overshoots samples beyond the valid data distribution. In contrast, GMFlow can provide a well-

defined conditional distribution $q_{\boldsymbol{\theta}}(\boldsymbol{u}|\boldsymbol{x}_t, \boldsymbol{c})$. This allows us to formulate probabilistic guidance, a principled approach that reweights the predicted distribution while preserving its intrinsic bounds and structure.

To reweight the GM PDF $q_{\boldsymbol{\theta}}(\boldsymbol{u}|\boldsymbol{x}_t, \boldsymbol{c})$ analytically, we multiply it by a Gaussian mask $w(\boldsymbol{u})$:

$$q_{\mathrm{w}}(\boldsymbol{u}|\boldsymbol{x}_t, \boldsymbol{c}) := \frac{w(\boldsymbol{u})}{Z}q_{\boldsymbol{\theta}}(\boldsymbol{u}|\boldsymbol{x}_t, \boldsymbol{c}), \quad (7)$$

where $Z$ is a normalization factor. The reweighted $q_{\mathrm{w}}(\boldsymbol{u}|\boldsymbol{x}_t, \boldsymbol{c})$ remains a GM with analytically derived parameters (see § D.2 for derivation), and is treated as the model output for sampling.

Then, our goal is to design $w(\boldsymbol{u})$ so that it enhances condition alignment without inducing OOD samples. To this end, we approximate the conditional and unconditional GM predictions $q_{\boldsymbol{\theta}}(\boldsymbol{u}|\boldsymbol{x}_t, \boldsymbol{c})$ and $q_{\boldsymbol{\theta}}(\boldsymbol{u}|\boldsymbol{x}_t)$ as isotropic Gaussian surrogates $\mathcal{N}(\boldsymbol{u}; \boldsymbol{\mu}_{\mathrm{c}}, s_{\mathrm{c}}^2\boldsymbol{I})$ and $\mathcal{N}(\boldsymbol{u}; \boldsymbol{\mu}_{\mathrm{u}}, s_{\mathrm{u}}^2\boldsymbol{I})$ by matching the mean and total variance of the GM (see § A.1 for details). Using these approximations, we define the unnormalized Gaussian mask as:

$$w(\boldsymbol{u}) := \frac{\mathcal{N}\big(\boldsymbol{u}; \boldsymbol{\mu}_{\mathrm{c}} + \tilde{w}s_{\mathrm{c}}\Delta\boldsymbol{\mu}_{\mathrm{n}}, \big(1 - \tilde{w}^2\big)s_{\mathrm{c}}^2\boldsymbol{I}\big)}{\mathcal{N}\big(\boldsymbol{u}; \boldsymbol{\mu}_{\mathrm{c}}, s_{\mathrm{c}}^2\boldsymbol{I}\big)}, \quad (8)$$

where $\tilde{w} \in [0, 1)$ is the probabilistic guidance scale, and $\Delta\boldsymbol{\mu}_{\mathrm{n}} := \frac{\boldsymbol{\mu}_{\mathrm{c}} - \boldsymbol{\mu}_{\mathrm{u}}}{\|\boldsymbol{\mu}_{\mathrm{c}} - \boldsymbol{\mu}_{\mathrm{u}}\|/\sqrt{D}}$ is the normalized mean difference.

Intuitively, the numerator in Eq. (8) shifts the conditional mean by the bias $\tilde{w}s_{\mathrm{c}}\Delta\boldsymbol{\mu}_{\mathrm{n}}$ to enhance conditioning, while reducing the conditional variance according to bias–variance decomposition. Notably, for any $\tilde{w}$, a sample $\boldsymbol{u}$ from the numerator Gaussian satisfies $\mathbb{E}_{\boldsymbol{u}}\big[\|\boldsymbol{u} - \boldsymbol{\mu}_{\mathrm{c}}\|^2/D\big] \equiv s_c^2$. This ensures that the original bounds of the conditional distribution are preserved.

Meanwhile, the denominator in Eq. (8) cancels out the original mean and variance of the conditional GM, so that the reweighted GM $q_{\mathrm{w}}(\boldsymbol{u}|\boldsymbol{x}_t, \boldsymbol{c})$ approximately inherits the adjusted mean and variance of the numerator. Since the "masking" operation retains the original GM components, the fine-grained structure of the conditional GM is preserved.

With preserved bounds and structure, probabilistic guidance reduces the risk of OOD samples compared to vanilla CFG, effectively preventing over-saturation and enhancing overall image quality.

### 3.3. GM-SDE and GM-ODE Solvers

In this subsection, we show that GMFlow enables unique SDE and ODE solvers that greatly reduce discretization errors by analytically deriving the reverse transition distribution and flow velocity field.

**GM-SDE solver.** Given the predicted $\boldsymbol{x}_0$-based GM

$q_{\boldsymbol{\theta}}(\boldsymbol{x}_0|\boldsymbol{x}_t) = \sum_k A_k \mathcal{N}(\boldsymbol{x}_0; \boldsymbol{\mu}_{\mathrm{x}k}, s_\mathrm{x}^2 \boldsymbol{I})$, the reverse transition distribution $q_{\theta}(\boldsymbol{x}_{t-\Delta t}|\boldsymbol{x}_t)$ can be analytically derived (see § C.3 for derivation):

$$q_{\boldsymbol{\theta}}(\boldsymbol{x}_{t-\Delta t}|\boldsymbol{x}_t)$$
$$= \sum_{k=1}^{K} A_k \mathcal{N}\left(\boldsymbol{x}_{t-\Delta t}; c_1 \boldsymbol{x}_t + c_2 \boldsymbol{\mu}_{\mathrm{x}k}, \left(c_3 + c_2^2 s_\mathrm{x}^2\right)\boldsymbol{I}\right), \quad (9)$$

with the coefficients $c_1 = \frac{\sigma_{t-\Delta t}^2}{\sigma_t^2}\frac{\alpha_t}{\alpha_{t-\Delta t}}$, $c_2 = \frac{\beta_{t,\Delta t}^2}{\sigma_t^2}\alpha_{t-\Delta t}$, $c_3 = \frac{\beta_{t,\Delta t}^2}{\sigma_t^2}\sigma_{t-\Delta t}^2$. By recursively sampling from $q_{\boldsymbol{\theta}}(\boldsymbol{x}_{t-\Delta t}|\boldsymbol{x}_t)$, we obtain a GM approximation to the reverse-time SDE solution. Alternatively, we can implement the solver by first sampling $\hat{\boldsymbol{x}}_0 \sim q_{\boldsymbol{\theta}}(\boldsymbol{x}_0|\boldsymbol{x}_t)$ and then sampling $p(\boldsymbol{x}_{t-\Delta t}|\boldsymbol{x}_t, \hat{\boldsymbol{x}}_0) = \mathcal{N}(\boldsymbol{x}_{t-\Delta t}; c_1 \boldsymbol{x}_t + c_2 \hat{\boldsymbol{x}}_0, c_3 \boldsymbol{I})$ (Song et al., 2021a) (see § C.3 for details). Theoretically, if $q_{\boldsymbol{\theta}}(\boldsymbol{x}_0|\boldsymbol{x}_t)$ is accurate, the GM-SDE solution incurs no error even in a single step. In practice, a smaller step size $\Delta t$ increases reliance on the mean (see § C.4 for details) over the shape of the distribution, which can be more accurate as per Theorem 3.1.

**GM-ODE solver.** While GMFlow supports standard ODE integration by converting its GM prediction into the current mean velocity, it also enables a unique sampling scheme with reduced discretization errors by analytically deriving the flow velocity field for any time $\tau < t$, thereby facilitating sub-step integration without additional neural network evaluations. Specifically, given the $\boldsymbol{x}_0$-based GM $q_{\boldsymbol{\theta}}(\boldsymbol{x}_0|\boldsymbol{x}_t)$ at $\boldsymbol{x}_t$, we show in § C.5 that the denoising distribution at $\boldsymbol{x}_\tau$ can be derived as:

$$\hat{q}(\boldsymbol{x}_0|\boldsymbol{x}_\tau) = \frac{p(\boldsymbol{x}_\tau|\boldsymbol{x}_0)}{Z \cdot p(\boldsymbol{x}_t|\boldsymbol{x}_0)} q_{\boldsymbol{\theta}}(\boldsymbol{x}_0|\boldsymbol{x}_t), \quad (10)$$

where $Z$ is a normalization factor, and $\hat{q}(\boldsymbol{x}_0|\boldsymbol{x}_\tau)$ is also a GM with analytically derived parameters. This allows us to instantly estimate GMFlow's next-step prediction at $\boldsymbol{x}_\tau$ from its current prediction at $\boldsymbol{x}_t$, without re-evaluating the neural network. The velocity can therefore be derived by reparameterizing $\hat{q}(\boldsymbol{x}_0|\boldsymbol{x}_\tau)$ in terms of $\boldsymbol{u}$ and computing its mean. This enables the GM-ODE solver, which integrates a curved trajectory along the analytic velocity field via multiple Euler sub-steps between $t$ and $t - \Delta t$ (Algorithm 2 line 18–22). Theoretically, if $q_{\boldsymbol{\theta}}(\boldsymbol{x}_0|\boldsymbol{x}_t)$ is accurate, the GM-ODE solution also incurs no error. In practice, the velocity field is only locally accurate as $\tau \to t$ and $\boldsymbol{x}_\tau \to \boldsymbol{x}_t$, thus multiple network evaluations are still required.

**Second-order multistep GM solvers.** Vanilla diffusion models often use Adams–Bashforth-like second-order multistep solvers (Lu et al., 2023), which extrapolate the mean $\boldsymbol{x}_0$ predictions of the last two steps to estimate the next midpoint, and then apply an Euler update. Analogously, we extend this approach to GMs. Given GM predictions at times $t$ and $t + \Delta t$, we first convert the latter to time $t$ via

Eq. (10), yielding $\hat{q}(\boldsymbol{x}_0|\boldsymbol{x}_t)$. In an ideal scenario where both GMs are accurate, $\hat{q}(\boldsymbol{x}_0|\boldsymbol{x}_t)$ perfectly matches $q_{\boldsymbol{\theta}}(\boldsymbol{x}_0|\boldsymbol{x}_t)$; otherwise, their discrepancy is extrapolated following the Adams–Bashforth scheme. To perform GM extrapolation, we adopt a GM reweighting scheme similar to that in § 3.2. More details are provided in § A.2.

### 3.4. Practical Designs

**Pixel-wise factorization.** For high-dimensional data such as images, Gaussian mixture models often suffer from mode collapse. To address this, we treat each pixel (in the latent grid for latent diffusion (Rombach et al., 2022)) as an independent low-dimensional GM, making the training loss the sum of per-pixel GM KL terms.

**Spectral sampling.** Due to the factorization, image generation under GM-SDE solvers performs the sampling step $\hat{\boldsymbol{x}}_0 \sim q_{\boldsymbol{\theta}}(\boldsymbol{x}_0|\boldsymbol{x}_t)$ independently for each pixel, neglecting spatial correlations. To address this, we adopt a spectral sampling technique that imposes correlations through the frequency domain. Specifically, we generate a spatially correlated Gaussian random field from a learnable power spectrum. The per-pixel Gaussian samples are then mapped to GM samples via Knothe–Rosenblatt transport (Knothe, 1957; Rosenblatt, 1952). More details are provided in § A.3.

**Transition loss.** Empirically, we observe that the GM KL loss may still induce minor gradient spikes when $s$ becomes small. To stabilize training, we adopt a modified loss based on the transition distributions (Eq. (9)), formulated as $\mathcal{L}_{\mathrm{trans}} = \mathbb{E}_{t,\boldsymbol{x}_{t-\Delta t},\boldsymbol{x}_t}[-\log q_{\boldsymbol{\theta}}(\boldsymbol{x}_{t-\Delta t}|\boldsymbol{x}_t)]$, where we define $\Delta t := \lambda t$ with the transition ratio hyperparameter $\lambda \in (0, 1]$. When $\lambda = 1$, the loss is equivalent to the original; when $\lambda < 1$, the transition distribution has a lower bound of variance $c_3$, which improves training stability. The modified training scheme is described in Algorithm 1.

## 4. Experiments

To evaluate the effectiveness of GMFlow, we compare it against vanilla flow matching baselines on two datasets: (a) a simple 2D checkerboard distribution, which facilitates visualization of sample histograms and analysis of underlying mechanisms; (b) the class-conditioned ImageNet dataset (Deng et al., 2009), a challenging benchmark that demonstrates the practical advantages of GMFlow for large-scale image generation.

### 4.1. Sampling a 2D Checkerboard Distribution

In this subsection, we compare GMFlow with the vanilla flow model on a simple 2D checkerboard distribution, following the experimental settings of Lipman et al. (2023). All configurations adopt the same 5-layer MLP architecture for denoising 2D coordinates, differing only in their output

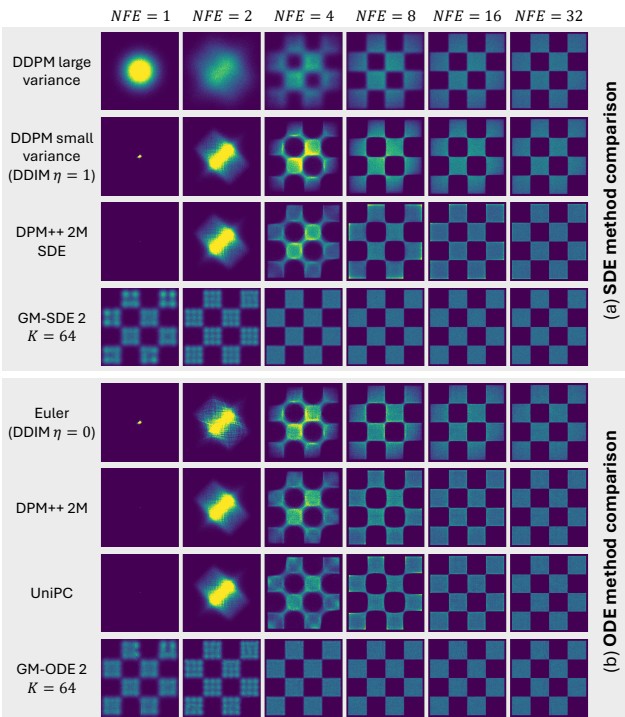

Figure 2. Comparison among vanilla flow models with different solvers and GMFlow. For both SDE and ODE, our method achieves higher quality in few-step sampling.

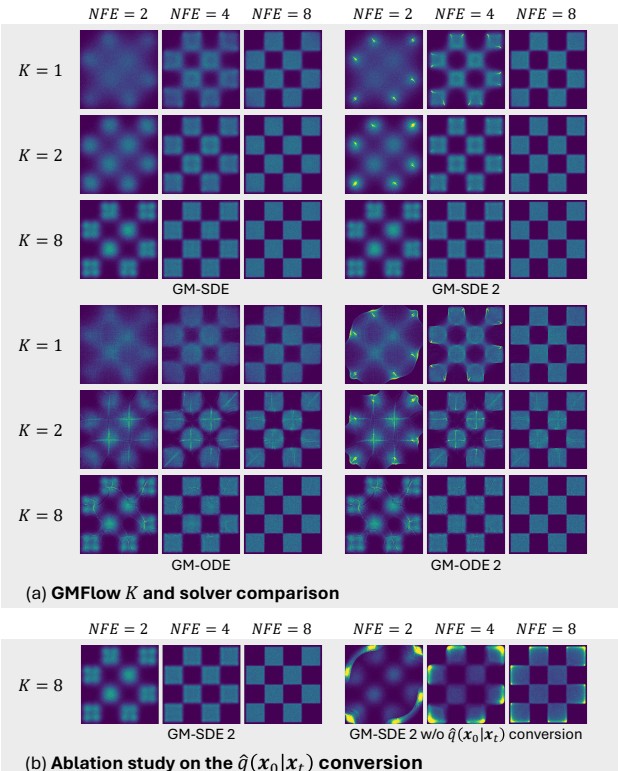

(a) **GMFlow $K$ and solver comparison**

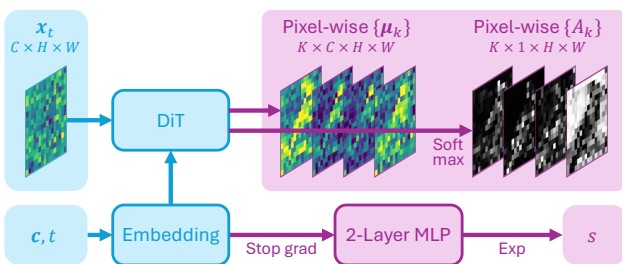

(b) **Ablation study on the $\hat{q}(\boldsymbol{x}_0|\boldsymbol{x}_t)$ conversion**

Figure 3. (a) Comparison of first- and second-order GM-SDE and GM-ODE solvers with varying $NFE$ and GM components $K$. Increasing $K$ improves few-step sampling results. Second-order solvers produce sharper histograms with fewer outliers than first-order solvers. (b) Ablation study on the $\hat{q}(\boldsymbol{x}_0|\boldsymbol{x}_t)$ conversion in second-order solvers. Removing this conversion causes samples to overshoot and concentrate at the edges.

channels. For GMFlow, we train multiple models with different numbers of GM components $K$ (with $\lambda = 0.9$); for GM-ODE sampling, we use $n = \lceil 128/NFE \rceil$ sub-steps.

**Comparison against flow model baseline.** In Fig. 2, we compare the 2D sample histograms of GMFlow using second-order GM solvers (GM-SDE/ODE 2) against the vanilla flow matching baseline using established SDE and ODE solvers (Lu et al., 2023; Zhao et al., 2023; Ho et al., 2020; Song et al., 2021a). Notably, vanilla flow models require approximately 8 steps to achieve a reasonable histogram and 16–32 steps for high-quality sampling. In contrast, GMFlow ($K = 64$) can approximate the checkerboard in a single step and achieve high-quality sampling in 4 steps. Moreover, for vanilla flow models, samples generated by first-order solvers (DDPM, Euler) tend to concentrate toward the center, whereas those from second-order solvers (DPM++, UniPC) concentrate near the outer edges. In contrast, GMFlow samples are highly uniform. This validates GMFlow's advantage in few-step sampling and multi-modal distribution modeling.

**Impact of the GM component count $K$.** Fig. 3 (a) demonstrates that increasing $K$ significantly improves few-step sampling results, leading to sharper histograms. Notably, GM-SDE sampling with $K = 1$ is theoretically equivalent to DDPM with learned variance (Nichol & Dhariwal, 2021), which produces a blurrier histogram. This further highlights

the expressiveness of GMFlow.

**Second-order GM solvers.** Fig. 3 (a) also compares our first- and second-order GM solvers across various NFEs and GM component counts. Comparing the left (first-order) and right (second-order) columns, we observe that second-order solvers produce sharper histograms and better suppress outliers, particularly when $NFE$ and $K$ are small. With $K = 8$, the GM probabilities are sufficiently accu-

Figure 4. Architecture of GMFlow-DiT. The original DiT (Peebles & Xie, 2023) is shown in blue, and the modified output layers are shown in purple.

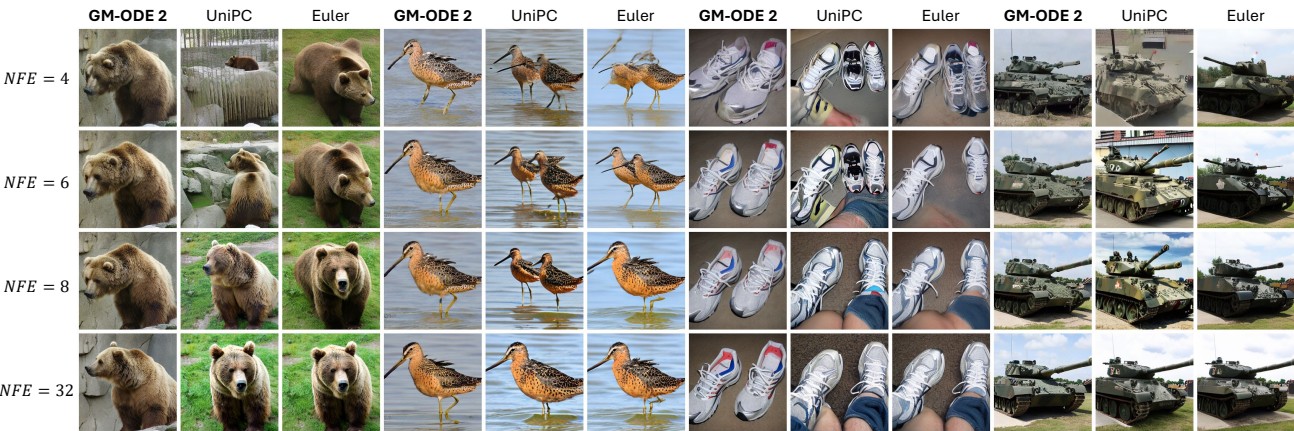

*Figure 5.* Qualitative comparisons (at best Precision) among GMFlow (GM-ODE 2) and vanilla flow model baselines (UniPC and Euler). GMFlow produces consistent results across various NFEs, whereas baselines struggle in few-step sampling, exhibiting distorted structures.

rate that the second-order solvers do not make a difference, aligning with our theoretical analysis.

**Ablation studies.** To validate the importance of the $\hat{q}(\boldsymbol{x}_0|\boldsymbol{x}_t)$ conversion in second-order GM solvers, we conduct an ablation study by removing the conversion and directly extrapolating the $\boldsymbol{x}_0$ distributions of the last two steps, similar to DPM++ (Lu et al., 2023). As shown in Fig. 3 (b), removing this conversion introduces an overshooting bias, similar to DPM++ 2M SDE ($NFE = 8$), underscoring its importance in maintaining sample uniformity.

### 4.2. ImageNet Generation

For image generation evaluation, we benchmark GMFlow against vanilla flow baselines on class-conditioned ImageNet 256×256. We train a Diffusion Transformer (DiT-XL/2) (Peebles & Xie, 2023; Vaswani et al., 2017) using the flow matching objective (Eq. (3)) as the baseline, and then adapt it into GMFlow-DiT by expanding its output channels and training it with the transition loss ($\lambda = 0.5$). As illustrated in Fig. 4, GMFlow-DiT produces $K$ weight maps and mean maps as the pixel-wise parameters $\{A_k, \boldsymbol{\mu}_k\}$. Meanwhile, a tiny two-layer MLP separately predicts $s$ using only the time $t$ and condition $\boldsymbol{c}$ as inputs. Empirically, we find this design to be more stable than predicting $s$ using the main DiT. The adaptation results in only a 0.2% increase in network parameters for $K = 8$. During inference, we apply the orthogonal projection technique by Sadat et al. (2025) to both the baseline and GMFlow since it universally improves Precision. Additional training and inference details are provided in § A.4.

**Evaluation protocol.** For quantitative evaluation, we adopt the standard metrics used in ADM (Dhariwal & Nichol, 2021), including Fréchet Inception Distance (FID) (Heusel et al., 2017), Inception Score (IS), and Precision–Recall (Kynkäänniemi et al., 2019). These metrics are highly sen-

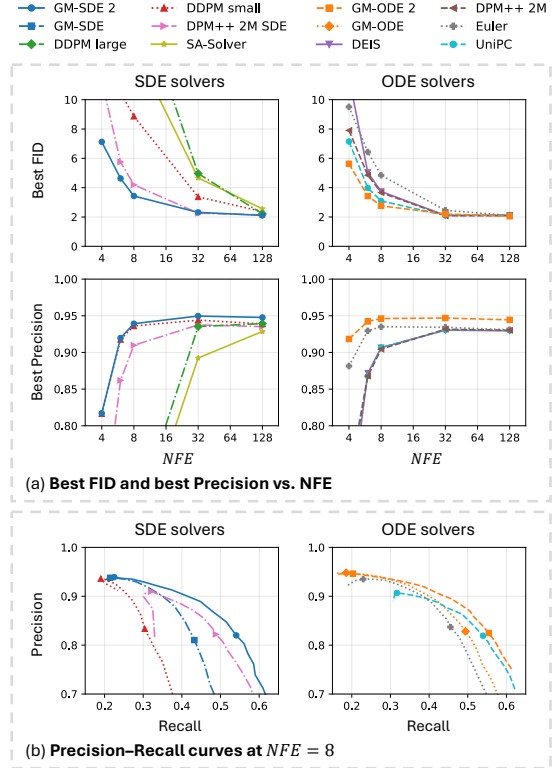

*Figure 6.* (a) Comparison of the best Precision and best FID among GMFlow and vanilla flow model baselines using different solvers across varying NFEs on ImageNet. For best FID, GMFlow significantly outperforms the baselines in few-step sampling; for best Precision, GMFlow consistently excels across different NFEs. (b) Precision-Recall curves of different methods at $NFE = 8$. Points corresponding to the best FID and best Precision are marked on the curves. GMFlow achieves superior Precision and Recall.

sitive to the classifier-free guidance (CFG) scale: while smaller CFG scales ($w \approx 1.4$) often yield the best FID by balancing diversity and quality, human users generally favor

higher CFG scales ($w > 3.0$) for the best perceptual quality, which also leads to the best Precision. The best Recall and IS values typically occur outside these CFG ranges and are less representative of typical usage. Therefore, for a fair and complete evaluation, we sweep over the CFG scale $w$ or the probabilistic guidance scale $\tilde{w}$ for each model, and report the best FID and best Precision. We also present Precision–Recall curves, illustrating the quality–diversity trade-off more comprehensively. Additionally, following Sadat et al. (2025), we report the Saturation metric at the best Precision setting to assess over-saturation.

**Comparison against flow model baselines.** For baselines, we test the vanilla flow model using various first-order solvers (DDPM (Ho et al., 2020), Euler (DDIM) (Song et al., 2021a)) and advanced second-order solvers (DPM++ (Lu et al., 2023), DEIS (Zhang & Chen, 2023), UniPC (Zhao et al., 2023), SA-Solver (Xue et al., 2023)). Details on adapting these solvers for flow matching are presented in § A.5. In Fig. 6 (a), we compare these baselines with our GMFlow ($K = 8$) model equipped with second-order GM solvers (GM-SDE/ODE 2). GMFlow consistently achieves superior Precision in both SDE and ODE sampling across various NFEs. Notably, GM-ODE 2 reaches a Precision of 0.942 in just 6 steps, while GM-SDE 2 attains a state-of-the-art Precision of 0.950 in 32 steps. For FID, GMFlow outperforms baselines in few-step settings ($NFE \leq 8$) and remains competitive in many-step settings ($NFE \geq 32$). Fig. 6 (b) further illustrates that the Precision-Recall curves of first- and second-order GM solvers consistently outperform those of their baseline counterparts. Qualitative comparisons are presented in Fig. 5.

**Saturation assessment.** Table 2 compares the Saturation metrics of different methods at their best Precision. GMFlow ($K = 8$) effectively reduces over-saturation, achieving Saturation levels closest to real data. Visual comparisons in Fig. 5 further support this finding. Additionally, ablating the orthogonal projection technique results in the same trend: GMFlow consistently achieves the best Saturation, while baseline methods perform even worse without orthogonal projection.

**Impact of the GM component count $K$.** In Fig. 7, we analyze the impact of the number of GM components on the metrics. Few-step sampling ($NFE = 8$) benefits the most from increasing GM components, particularly with the GM-SDE 2 solver. The metrics generally saturate at $K = 8$. Beyond this point, the GM-SDE 2 Precisions decline because spectral sampling introduces larger numerical errors when more Gaussian components are used. Additionally, Table 3 presents the time-averaged negative log-likelihood (NLL) values of data under the predicted distribution $q_{\boldsymbol{\theta}}(\boldsymbol{x}_0|\boldsymbol{x}_t, \boldsymbol{c})$, showing that increasing the number of Gaussians significantly reduces NLL, suggesting its potential benefits for

*Table 2.* ImageNet evaluation results at best Precision ($NFE = 32$). The reported Saturation values (Sadat et al., 2025) are relative to the real data statistics (Saturation=0.318).

| Method | Orthogonal projection | Guidance | Precision↑ | Saturation |
|---|---|---|---|---|
| DDPM small | ✓ | $w = 3.3$ | 0.944 | +0.032 |
| DPM++ 2M SDE | ✓ | $w = 2.9$ | 0.938 | +0.032 |
| **GM-SDE** | ✓ | $\tilde{w} = 0.47$ | 0.950 | −0.019 |
| **GM-SDE 2** | ✓ | $\tilde{w} = 0.39$ | 0.950 | −0.019 |
| Euler | ✓ | $w = 3.3$ | 0.934 | +0.052 |
| UniPC | ✓ | $w = 3.3$ | 0.931 | +0.048 |
| **GM-ODE** | ✓ | $\tilde{w} = 0.47$ | 0.947 | −0.024 |
| **GM-ODE 2** | ✓ | $\tilde{w} = 0.47$ | 0.947 | −0.024 |
| DDPM small | | $w = 3.3$ | 0.939 | +0.056 |
| DPM++ 2M SDE | | $w = 2.9$ | 0.933 | +0.054 |
| **GM-SDE** | | $\tilde{w} = 0.27$ | 0.943 | +0.023 |
| Euler | | $w = 3.3$ | 0.931 | +0.064 |
| UniPC | | $w = 3.3$ | 0.929 | +0.060 |
| **GM-ODE** | | $\tilde{w} = 0.27$ | 0.933 | +0.016 |

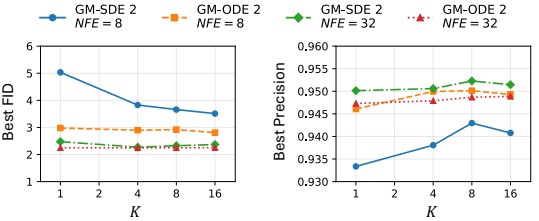

*Figure 7.* Best FIDs and best Precisions of GMFlow models with varying numbers of GM components $K$.

*Table 3.* Validation NLL of ImageNet training images (latents) under different GMFlow configurations.

| Method | $K = 1$ | $K = 4$ | $K = 8$ | $K = 16$ | $K = 16$ +spectral |
|---|---|---|---|---|---|
| **NLL (bits/dim)** | 0.346 | 0.263 | 0.242 | 0.224 | 0.173 |

posterior sampling applications.

**Ablation studies.** Table 4 presents ablation study results evaluating the impact of various design choices in our method. For GM-SDE, removing spectrum sampling (A2) leads to a noticeable degradation in FID. Replacing our probabilistic guidance with vanilla CFG (A3)—i.e., naively shifting the mean of the predicted GM—results in severe quality degradation (lower Precision) and over-saturation, which is also evident in Fig. 8. Replacing GM-SDE 2 with the second-order DPM++ SDE solver (A4) worsens both FID and Precision. Reducing the transition loss to the original KL loss (A5) degrades FID. Finally, for GM-ODE 2, directly applying ODE integration without taking sub-steps (B1) leads to significantly worse FID.

**Inference time.** GMFlow only alters the output layer and uses solvers based on simple arithmetic operations. As a result, it adds only 0.005 sec of overhead per step (batch size 125, A100 GPU) compared to its flow-matching counterpart,

which is minimal compared to the total inference time of 0.39 sec per step—most of which is spent on DiT.

## 5. Related Work

Prior works (Nichol & Dhariwal, 2021; Bao et al., 2022b;a) extend standard diffusion models by learning the variance of denoising distributions, which is effectively a special case of GMFlow with $K = 1$ and learnable $s$. GMS (Guo et al., 2023) further extends this approach to third-order moments and fits a bimodal GM to the moments during inference. In § B.3, we present a comparison between GMFlow ($K = 2$) and GMS. Conversely, Xiao et al. (2022) employ a generative adversarial network (Goodfellow et al., 2014) to capture a more expressive distribution, though adversarial diffusion typically relies on additional objectives for better quality and stability (Jolicoeur-Martineau et al., 2021; Kim et al., 2024). However, none of these methods address deterministic ODE sampling.

A growing line of work trains networks to directly predict ODE integrals or denoised samples, enabling extremely few-step sampling (Salimans & Ho, 2022; Song et al., 2023; Yin et al., 2024b; Sauer et al., 2024). These approaches fall under the category of distillation methods, whose quality typically remains bounded by the many-step performance of standard diffusion models, unless supplemented by additional adversarial training (Kim et al., 2024; Yin et al., 2024a).

Efforts to refine CFG beyond thresholding (Saharia et al., 2022a) and orthogonal projection (Sadat et al., 2025) include disabling CFG in early steps (Kynkäänniemi et al., 2024), which compromises Precision, and adding Langevin corrections (Bradley & Nakkiran, 2024), which reduces efficiency.

Beyond diffusion models, Gaussian mixtures have also been employed in other generative models. DeLiGAN (Gurumurthy et al., 2017) and GM-GAN (Ben-Yosef & Weinshall, 2018) enhance the latent expressiveness of GANs through the introduction of mixture priors. GIVT (Tschannen et al., 2024) models the output of an autoregressive Transformer as a Gaussian mixture distribution, enabling continuous data sampling and outperforming its quantization-based counterpart.

## 6. Conclusion

In this work, we introduced GMFlow, a generalization of diffusion and flow models that represents flow velocity as a Gaussian mixture, offering greater expressiveness for capturing complex multi-modal structures. We derived principled SDE/ODE solvers unique to this approach and proposed a novel probabilistic guidance technique to eliminate

*Table 4.* Ablation studies on GMFlow ($K = 8$, $NFE = 8$) using ImageNet evaluation.

| ID | Method | Best FID↓ | Best Precision↑ | Saturation @Best Prec. |
|---|---|---|---|---|
| **A0** | **Full model (GM-SDE 2)** | 3.43 | 0.939 | −0.003 |
| A1 | A0 → GM-SDE | 6.96 (+3.53) | 0.938 (−0.001) | +0.009 |
| A2 | A1 w/o spec. sampling | 8.98 (+2.02) | 0.940 (+0.002) | −0.022 |
| A3 | A2 → Vanilla CFG | 9.02 (+0.04) | 0.917 (−0.023) | +0.049 |
| A4 | A0 → DPM++ 2M SDE | 4.59 (+1.16) | 0.912 (−0.027) | −0.062 |
| A5 | A0 → $\lambda = 1.0$ | 4.49 (+1.06) | 0.941 (+0.002) | −0.019 |
| **B0** | **Full model (GM-ODE 2)** | 2.77 | 0.946 | −0.028 |
| B1 | B0 w/o sub-steps | 7.47 (+4.70) | 0.947 (+0.001) | −0.031 |

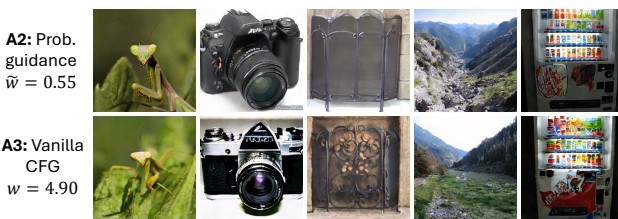

**A2:** Prob. guidance $\widetilde{w} = 0.55$

**A3:** Vanilla CFG $w = 4.90$

*Figure 8.* Ablation study on probabilistic guidance, comparing samples from Table 4 A2 and A3.

over-saturation. Extensive experiments demonstrated that GMFlow significantly improves few-step generation while enhancing overall sample quality. This framework lays the foundation for potential future research in both theoretical and practical directions, including applications such as posterior sampling with GMFlow priors.

## Limitations

To apply Gaussian mixture to high-dimensional data, we adopted pixel-wise factorization for image generation, which may not fully exploit the potential of GMFlow, leaving rooms for further development.

## Acknowledgments

We thank Liwen Wu, Lvmin Zhang, and Guandao Yang for discussions and feedback. Part of this work was done while Hansheng Chen was an intern at Adobe Research. This project was partially supported by Qualcomm Innovation Fellowship, ARL grant W911NF-21-2-0104, and Vannevar Bush Faculty Fellowship.

## Impact Statement

This paper presents work whose goal is to advance the field of generation models. There are many potential societal consequences of our work, none which we feel must be specifically highlighted here.

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

# A. Additional Technical Details

## A.1. Details on Gaussian Surrogates

For a GM distribution of the form $\sum_{k=1}^{K} A_k \mathcal{N}(\boldsymbol{\mu}_k, s^2\boldsymbol{I})$, we approximate it with an isotropic Gaussian surrogate $\mathcal{N}(\boldsymbol{\mu}', s'^2\boldsymbol{I})$ by matching the first two moments. The surrogate mean is computed as:

$$\boldsymbol{\mu}' = \sum_{k=1}^{K} A_k \boldsymbol{\mu}_k. \tag{11}$$

To determine the surrogate variance, we equate the trace of the GM's covariance matrix (i.e., the GM's total variance) with the trace of the surrogate covariance, which yields:

$$s'^2 = \frac{1}{D} \sum_{k=1}^{K} A_k \|\boldsymbol{\mu}_k - \boldsymbol{\mu}'\|^2 + s^2. \tag{12}$$

## A.2. Details on Second-Order GM Solvers

For GM extrapolation, we follow the method in § 3.2 and fit the isotropic Gaussian surrogates $\mathcal{N}(\boldsymbol{x}_0; \boldsymbol{\mu}_+, s_+^2\boldsymbol{I}) :\approx q_{\boldsymbol{\theta}}(\boldsymbol{x}_0|\boldsymbol{x}_t)$, $\mathcal{N}(\boldsymbol{x}_0; \boldsymbol{\mu}_-, s_-^2\boldsymbol{I}) :\approx \hat{q}(\boldsymbol{x}_0|\boldsymbol{x}_t)$. We then define a Gaussian mask:

$$\rho(\boldsymbol{x}_0) = \frac{\mathcal{N}\left(\boldsymbol{x}_0; \boldsymbol{\mu}_+ + \Delta\boldsymbol{\mu}, \left(s_+^2 - \frac{\|\Delta\boldsymbol{\mu}\|^2}{D}\right)\boldsymbol{I}\right)}{\mathcal{N}(\boldsymbol{x}_0; \boldsymbol{\mu}_+, s_+^2\boldsymbol{I})}, \tag{13}$$

where $\Delta\boldsymbol{\mu} = \frac{\boldsymbol{\mu}_+ - \boldsymbol{\mu}_-}{2}$, so that $\boldsymbol{\mu}_+ + \Delta\boldsymbol{\mu}$ represents the extrapolated mean at the next midpoint. The final extrapolated GM is obtained via the reweighting formulation:

$$q_{\text{ext}}(\boldsymbol{x}_0|\boldsymbol{x}_t) = \frac{\rho(\boldsymbol{x}_0)}{Z} q_{\boldsymbol{\theta}}(\boldsymbol{x}_0|\boldsymbol{x}_t). \tag{14}$$

We then feed $q_{\text{ext}}(\boldsymbol{x}_0|\boldsymbol{x}_t)$ to the first-order GM-SDE or ODE solver as a substitution for $q_{\boldsymbol{\theta}}(\boldsymbol{x}_0|\boldsymbol{x}_t)$.

Empirically, we observe that second-order GM solvers using the above formulation slightly underperform with higher guidance scales, which is a common issue with multistep solvers (Lu et al., 2023; Zhao et al., 2023). To address this, we rescale the mean difference $\Delta\boldsymbol{\mu}$ by an empirical factor $\sqrt{\max(0, 1 - \frac{(\tilde{w}^2 + c_a)s_c^2}{\Delta t^2})}$ (with the hyperparameter $c_a = 0.005$), so that a high $\tilde{w}$ practically disables multistep extrapolation.

## A.3. Details on Spectral Sampling

Due to pixel-wise factorization, image generation under GM-SDE solvers performs the sampling step $\hat{\boldsymbol{x}}_0 \sim q_{\boldsymbol{\theta}}(\boldsymbol{x}_0|\boldsymbol{x}_t)$ independently for each pixel, neglecting spatial correlations. Spectral sampling addresses this by establishing an invertible mapping between a frequency-space

---

**Algorithm 1:** Complete GMFlow training scheme.

**Input:** Data distribution $p(\boldsymbol{x}_0, \boldsymbol{c})$, transition ratio $\lambda$
**Output:** Network params $\boldsymbol{\theta}$
1   Initialize network params $\boldsymbol{\theta}$
2   **for** *sample* $\{\boldsymbol{x}_0, \boldsymbol{c}\} \sim p(\boldsymbol{x}_0, \boldsymbol{c})$ **do**
3     **if** *use logit-normal* **then**
4       Sample $t \sim LogitNormal(0, 1)$
5     **else**
6       Sample $t \sim U(0, 1)$
7     $\Delta t \leftarrow \lambda t$
8     Sample $\boldsymbol{x}_{t-\Delta t} \sim p(\boldsymbol{x}_{t-\Delta t}|\boldsymbol{x}_0)$
9     Sample $\boldsymbol{x}_t \sim p(\boldsymbol{x}_t|\boldsymbol{x}_{t-\Delta t})$
10    Predict GM params in $q_{\boldsymbol{\theta}}(\boldsymbol{x}_{t-\Delta t}|\boldsymbol{x}_t, \boldsymbol{c})$    // Eq. (9)
11    $\mathcal{L}_{\text{trans}} \leftarrow -\log q_{\boldsymbol{\theta}}(\boldsymbol{x}_{t-\Delta t}|\boldsymbol{x}_t, \boldsymbol{c})$
12    Predict magnitude spectrum $\boldsymbol{s}_\text{F}$
13    $\mathcal{L}_{\text{spec}} \leftarrow -\log \mathcal{N}\left(\text{vec}(\boldsymbol{z}_\text{r}); \boldsymbol{0}, \text{diag}(\boldsymbol{s}_\text{F})^2\right)$   // Eq. (20)
14    Backpropagate and update $\boldsymbol{\theta}$

---

**Algorithm 2:** Complete GMFlow sampling scheme.

**Input:** Steps $NFE$, sub-steps $n$, guidance scale $\tilde{w}$, condition $\boldsymbol{c}$, network params $\boldsymbol{\theta}$
**Output:** $\boldsymbol{x}_0$
1   $t \leftarrow 1, \Delta t = \frac{1}{NFE}, \boldsymbol{x}_1 \sim \mathcal{N}(\boldsymbol{0}, \boldsymbol{I}), Cache \leftarrow \{\}$
2   **while** $t > 0$ **do**
3     Predict GM params in $q_{\boldsymbol{\theta}}(\boldsymbol{x}_0|\boldsymbol{x}_t, \boldsymbol{c})$
4     **if** $\tilde{w} > 0$ **then**
5       Predict GM params in $q_{\boldsymbol{\theta}}(\boldsymbol{x}_0|\boldsymbol{x}_t)$
6       Compute $q_\text{w}(\boldsymbol{x}_0|\boldsymbol{x}_t, \boldsymbol{c})$    // Eq. (7)
7       $q_{\boldsymbol{\theta}}(\boldsymbol{x}_0|\boldsymbol{x}_t, \boldsymbol{c}) \overset{\text{param}}{\leftarrow} q_\text{w}(\boldsymbol{x}_0|\boldsymbol{x}_t, \boldsymbol{c})$
8     **if** *use 2nd-order* **then**
9       **if** $Cache \neq \{\}$ **then**
10         Compute $\hat{q}(\boldsymbol{x}_0|\boldsymbol{x}_t, \boldsymbol{c})$ from $Cache$   // Eq. (10)
11         $Cache \leftarrow \{\boldsymbol{x}_t, q_{\boldsymbol{\theta}}(\boldsymbol{x}_0|\boldsymbol{x}_t, \boldsymbol{c})\}$
12         Compute $q_{\text{ext}}(\boldsymbol{x}_0|\boldsymbol{x}_t, \boldsymbol{c})$    // Eq. (14)
13         $q_{\boldsymbol{\theta}}(\boldsymbol{x}_0|\boldsymbol{x}_t, \boldsymbol{c}) \overset{\text{param}}{\leftarrow} q_{\text{ext}}(\boldsymbol{x}_0|\boldsymbol{x}_t, \boldsymbol{c})$
14       **else**
15         $Cache \leftarrow \{\boldsymbol{x}_t, q_{\boldsymbol{\theta}}(\boldsymbol{x}_0|\boldsymbol{x}_t, \boldsymbol{c})\}$
16     **if** *use GM-SDE* **then**
17       **if** *use spectral sampling* **then**
18         Predict magnitude spectrum $\boldsymbol{s}_\text{F}$
19         Sample $\hat{\boldsymbol{x}}_0 \sim q_{\boldsymbol{\theta}}^\text{S}(\boldsymbol{x}_0|\boldsymbol{x}_t, \boldsymbol{c})$   // Eq. (19)
20       **else**
21         Sample $\hat{\boldsymbol{x}}_0 \sim q_{\boldsymbol{\theta}}(\boldsymbol{x}_0|\boldsymbol{x}_t, \boldsymbol{c})$
22       Sample $\boldsymbol{x}_{t-\Delta t} \sim \mathcal{N}(\boldsymbol{x}_{t-\Delta t}; c_1\boldsymbol{x}_t + c_2\hat{\boldsymbol{x}}_0, c_3\boldsymbol{I})$
23     **else if** *use GM-ODE* **then**
24       $\tau \leftarrow t, h = \Delta t/n$
25       **while** $\tau > t - \Delta t$ **do**
26         Compute $\hat{q}(\boldsymbol{x}_0|\boldsymbol{x}_\tau, \boldsymbol{c})$    // Eq. (10)
27         $\boldsymbol{x}_{\tau-h} = \boldsymbol{x}_\tau - h\mathbb{E}_{\boldsymbol{x}_0 \sim \hat{q}(\boldsymbol{x}_0|\boldsymbol{x}_\tau, \boldsymbol{c})}\left[\frac{\boldsymbol{x}_\tau - \boldsymbol{x}_0}{\sigma_\tau}\right]$
28         $\tau \leftarrow \tau - h$
29     $t \leftarrow t - \Delta t$

---

distribution (power spectrum) and the pixel-space GM distribution. During training, the power spectrum is optimized via

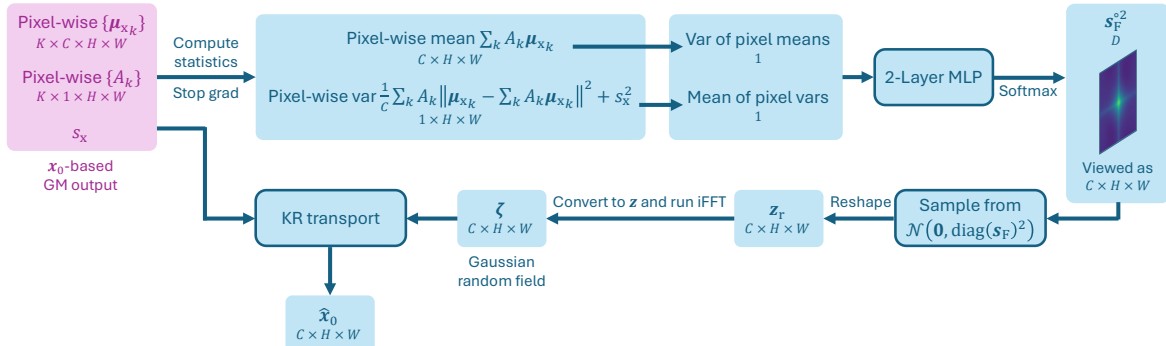

*Figure 9.* The optional spectral sampling pipeline used during inference. The spectrum MLP takes the statistics of pixel-wise GMs as input, and predicts the power spectrum $s_{\mathrm{F}}^{\circ 2}$. Alternatively, if probabilistic guidance or second-order GM solvers are employed, one can re-use the mean and variance from the numerator in the Gaussian mask to compute the input statistics, which serves as a good approximation.

a likelihood loss, while during inference, frequency-space samples are transformed back into pixel space to introduce spatial correlation.

We establish an invertible mapping using two building blocks: the fast Fourier transform (FFT) and Knothe–Rosenblatt (KR) transport (Knothe, 1957; Rosenblatt, 1952). The FFT bridges a frequency-space Gaussian distribution and a pixel-space Gaussian distribution, while KR transport bridges each per-pixel Gaussian to its corresponding Gaussian mixture (GM) distribution.

During training, given a real image $\boldsymbol{x}_0 \in \mathbb{R}^{C \times H \times W}$ and the factorized denoising distribution $q_{\boldsymbol{\theta}}(\boldsymbol{x}_0|\boldsymbol{x}_t) = \prod_{i=1,j=1}^{H,W} \sum_{k=1}^{K} A_{k,ij}\mathcal{N}(\boldsymbol{x}_{0,ij}|\boldsymbol{\mu}_{k,ij}, s^2\boldsymbol{I})$, KR transport defines an invertible mapping for each pixel:

$$T_{ij} : \boldsymbol{x}_{0,ij} \mapsto \boldsymbol{\zeta}_{ij}, \tag{15}$$

such that each $\boldsymbol{\zeta}_{ij}$ can be viewed as a standard Gaussian sample. We then assemble all $\boldsymbol{\zeta}_{ij}$ into a tensor $\boldsymbol{\zeta} \in \mathbb{R}^{C \times H \times W}$ and apply a forward 2D FFT with orthogonal normalization, yielding the complex frequency representation $\boldsymbol{z} = FFT(\boldsymbol{\zeta}) \in \mathbb{C}^{C \times H \times W}$. Since $\boldsymbol{z}$ is Hermitian symmetric, we can derive a real-valued representation $\boldsymbol{z}_{\mathrm{r}}$ while preserving invertibility:

$$\boldsymbol{z}_{\mathrm{r}} = \mathrm{Re}[\boldsymbol{z}] + \mathrm{Im}[\boldsymbol{z}]. \tag{16}$$

Finally, we impose a zero-mean Gaussian prior on $\boldsymbol{z}_{\mathrm{r}}$, given by $\mathcal{N}(\mathrm{vec}(\boldsymbol{z}_{\mathrm{r}}); \boldsymbol{0}, \mathrm{diag}(\boldsymbol{s}_{\mathrm{F}})^2)$, where $\boldsymbol{s}_{\mathrm{F}} \in \mathbb{R}_+^D$ represents the magnitude spectrum. To dynamically model $\boldsymbol{s}_{\mathrm{F}} \in \mathbb{R}_+^D$, we use a tiny two-layer MLP, which takes the mean of per-pixel GM variances and the variance of per-pixel GM means as input, and outputs the power spectrum $\boldsymbol{s}_{\mathrm{F}}^{\circ 2}$ with a softmax activation, where $(\cdot)^{\circ 2}$ stands for element-wise square. With this invertible mapping and the spectral Gaussian prior, we can derive the model's spectrum-enhanced denoising PDF using the change of variables technique in a similar way to normalizing flow models (Rezende & Mohamed, 2015),

which can be expressed as:

$$q_{\boldsymbol{\theta}}^{\mathrm{S}}(\boldsymbol{x}_0|\boldsymbol{x}_t) = \left|\det\left(\frac{\partial \mathrm{vec}(\boldsymbol{z}_{\mathrm{r}})}{\partial \boldsymbol{x}_0}\right)\right| \mathcal{N}(\mathrm{vec}(\boldsymbol{z}_{\mathrm{r}}); \boldsymbol{0}, \mathrm{diag}(\boldsymbol{s}_{\mathrm{F}})^2), \tag{17}$$

where the absolute determinant can be easily derived using the properties of FFT and KR transport:

$$\left|\det\left(\frac{\partial \mathrm{vec}(\boldsymbol{z}_{\mathrm{r}})}{\partial \boldsymbol{x}_0}\right)\right| = \left|\det\left(\frac{\partial \mathrm{vec}(\boldsymbol{z}_{\mathrm{r}})}{\partial \mathrm{vec}(\boldsymbol{\zeta})}\right)\right| \cdot \left|\det\left(\frac{\partial \mathrm{vec}(\boldsymbol{\zeta})}{\partial \boldsymbol{x}_0}\right)\right|$$
$$= 1 \cdot \frac{q_{\boldsymbol{\theta}}(\boldsymbol{x}_0|\boldsymbol{x}_t)}{\mathcal{N}(\mathrm{vec}(\boldsymbol{\zeta}); \boldsymbol{0}, \boldsymbol{I})}$$
$$= \frac{q_{\boldsymbol{\theta}}(\boldsymbol{x}_0|\boldsymbol{x}_t)}{\mathcal{N}(\mathrm{vec}(\boldsymbol{z}_{\mathrm{r}}); \boldsymbol{0}, \boldsymbol{I})}. \tag{18}$$

Substituting Eq. (18) into Eq. (17), we obtain:

$$q_{\boldsymbol{\theta}}^{\mathrm{S}}(\boldsymbol{x}_0|\boldsymbol{x}_t) = q_{\boldsymbol{\theta}}(\boldsymbol{x}_0|\boldsymbol{x}_t)\frac{\mathcal{N}(\mathrm{vec}(\boldsymbol{z}_{\mathrm{r}}); \boldsymbol{0}, \mathrm{diag}(\boldsymbol{s}_{\mathrm{F}})^2)}{\mathcal{N}(\mathrm{vec}(\boldsymbol{z}_{\mathrm{r}}); \boldsymbol{0}, \boldsymbol{I})}. \tag{19}$$

With the derived PDF, the entire model can be trained by minimizing the negative log-likelihood (NLL) of data samples under the PDF, which inherently includes the GM KL loss (Eq. (5) and (6)) as one of its terms. Therefore, the total loss consists of the GM KL loss (replaced with the transition loss in practice) and an additional spectral loss, defined as:

$$\mathcal{L}_{\mathrm{spec}} = \mathbb{E}_{t,\boldsymbol{x}_0,\boldsymbol{x}_t}\left[-\log\frac{\mathcal{N}(\mathrm{vec}(\boldsymbol{z}_{\mathrm{r}}); \boldsymbol{0}, \mathrm{diag}(\boldsymbol{s}_{\mathrm{F}})^2)}{\mathcal{N}(\mathrm{vec}(\boldsymbol{z}_{\mathrm{r}}); \boldsymbol{0}, \boldsymbol{I})}\right]$$
$$= \mathbb{E}_{t,\boldsymbol{x}_0,\boldsymbol{x}_t}\left[\frac{1}{2}\left\|\left(\mathrm{diag}(\boldsymbol{s}_{\mathrm{F}})^{-1} - \boldsymbol{I}\right)\mathrm{vec}(\boldsymbol{z}_{\mathrm{r}})\right\|^2 + \log\det(\mathrm{diag}(\boldsymbol{s}_{\mathrm{F}}))\right]. \tag{20}$$

In practice, we stop the gradient flow through KR transport to prevent spectral learning from influencing the main GMFlow model.

During inference, we employ the spectral sampling pipeline illustrated in Fig. 9.

## A.4. ImageNet Experiment Details.

We train both the baseline and GMFlow-DiT on ImageNet 256×256 with a batch size of 4096 images across 16 A100 GPUs, using a total training schedule of 200K iterations. We adopt the 8-bit AdamW (Dettmers et al., 2022; Loshchilov & Hutter, 2019) optimizer with a fixed learning rate of 0.0002. Following Stable Diffusion 3 (Esser et al., 2024), both models sample $t$ from a logit-normal distribution during training (Algorithm 1), which accelerates convergence.

While we set the transition ratio to $\lambda = 0.5$ in the main experiments, the results in Fig. 7 and Table 3 are based on an earlier design iteration that uses randomly sampled transition ratios $\lambda \sim LogitNormal(0, 1)$. This setting slightly increases Precision and decreases Recall, though the overall differences remain minor.

We densely evaluate the models across different guidance scales to identify the optimal FID and Precision. For vanilla CFG, we use guidance scales $w$ from the set $\{1.2, 1.3, 1.4, 1.5, 1.6, 1.7, 1.8, 1.9, 2.1, 2.3, 2.6, 2.9, 3.3, 3.7, 4.3, 4.9, 5.7, 6.5\}$; for probabilistic guidance, we use probabilistic guidance scales $\tilde{w}$ from the set $\{0.02, 0.03, 0.04, 0.05, 0.06, 0.07, 0.08, 0.09, 0.11, 0.13, 0.16, 0.19, 0.23, 0.27, 0.33, 0.39, 0.47, 0.55, 0.65, 0.75\}$. In practice (Algorithm 2), we implement probabilistic guidance on $\boldsymbol{x}_0$-based GMs, which is equivalent to guidance on $\boldsymbol{u}$.

For inference with GM-ODE solvers, we generally set the number of sub-steps to $n = \max\left(\frac{128}{NFE}, 2\right)$, which performs well when $NFE \geq 8$. For the exception when $NFE = 4$ or $6$, we observe that reducing $n$ to $8$ yields better performance.

In Table 3, the time-averaged NLL values are computed on 50K samples from the training dataset using the following equation:

$$NLL = \frac{1}{D}\mathbb{E}_{t,\boldsymbol{x}_0,\boldsymbol{x}_t}[-\log_2 q_{\boldsymbol{\theta}}(\boldsymbol{x}_0|\boldsymbol{x}_t)], \qquad (21)$$

where $t \sim U(0, 1)$ and $D = C \times H \times W$. This is equivalent to the original training loss ($\lambda = 1$) with a uniform time sampling scheme. When spectral prior is enabled, we replace $q_{\boldsymbol{\theta}}(\boldsymbol{x}_0|\boldsymbol{x}_t)$ with $q_{\boldsymbol{\theta}}^{\mathrm{S}}(\boldsymbol{x}_0|\boldsymbol{x}_t)$ (Eq. (19)).

### A.5. Adapting Diffusion Solvers for Flow Matching

For DDPM solvers (Ho et al., 2020), we implement them as special cases of GM-SDE solver with $K = 1$. The original DDPM solvers include a large variance variant ($\beta$) and a small variance variant ($\tilde{\beta}$). These are equivalent to setting $s = \frac{1}{\sqrt{\alpha_t^2 + \sigma_t^2}}$ and $s = 0$, respectively.

For DDIM solvers (Song et al., 2021a), its stochastic variant ($\eta = 1$) is equivalent to DDPM with small variance, whereas its deterministic variant ($\eta = 0$) is equivalent to Euler solver in flow matching models.

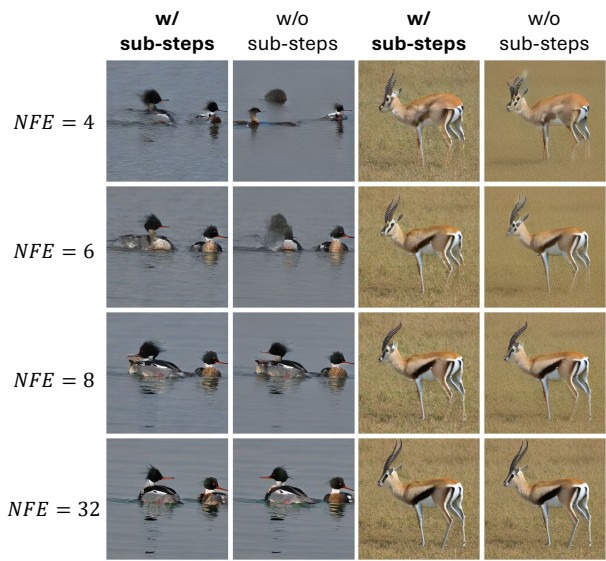

| w/ sub-steps | w/o sub-steps | w/ sub-steps | w/o sub-steps |

*Figure 10.* Qualitative comparison (at best Precision) from the ablation study on GM-ODE sub-steps. Without sub-steps (forcing $n = 1$), few-step sampling tends to produce over-smoothed textures and poor detail.

For DPM++ (Lu et al., 2023), DEIS (Zhang & Chen, 2023), UniPC (Zhao et al., 2023), and SA-Solver (Xue et al., 2023), we use their Diffusers implementations (von Platen et al., 2022) and rescale their noise schedules to match the flow matching noise schedule. This approach is similar to how the EDM Euler solver (Karras et al., 2022) rescales a variance-preserving diffusion model into a variance-exploding one at test time.

## B. Additional Experiment Results

### B.1. Ablation Study on GM-ODE Sub-Steps

In addition to the results in Table 4 (B0 and B1), Fig. 10 presents additional qualitative results from the ablation study on GM-ODE sub-steps, using $K = 8$ and the GM-ODE 2 solver. As shown in the figure, sub-steps are essential for producing detailed textures when $NFE < 8$.

### B.2. Additional Qualitative Comparison

Fig. 11 presents a comparison among uncurated random samples (conditioned on random class labels) from GM-Flow and vanilla flow matching baselines, showcasing their respective best results under the many-step setting. Overall, the images generated by GMFlow exhibit more natural color saturation and, in some cases, improved structural coherence. These observations are consistent with the quantitative results shown in Fig. 6 and Table 2.

*Table 5.* FIDs on CIFAR-10 unconditional image generation. Competitor results are sourced from Guo et al. (2023).

| NFE | 10 | 25 | 50 | 100 |
|---|---|---|---|---|
| DDPM large | 205.31 | 84.71 | 37.35 | 14.81 |
| DDPM small | 34.76 | 16.18 | 11.11 | 8.38 |
| SN-DDPM | 16.33 | 6.05 | 4.19 | 3.83 |
| GMS | 13.80 | 5.48 | 4.00 | 3.46 |
| **GM-SDE 2** | 9.11 | 4.16 | 3.79 | 3.76 |

### B.3. Comparison with Moment Matching Methods

Table 5 presents a quantitative comparison among GM-Flow ($K = 2$), GMS (Guo et al., 2023), SN-DDPM (Bao et al., 2022a), and DDPM (Ho et al., 2020) for CIFAR-10 (Krizhevsky et al., 2009) unconditional image generation using SDE sampling. The GMFlow model is trained from scratch using the same U-Net architecture (Ronneberger et al., 2015; Ho et al., 2020) as its competitors, but with modified output layers as in GM-DiT. The results demonstrate that GMFlow significantly outperforms its competitors in few-step sampling.

## C. Additional Theoretical Analysis

### C.1. Details on the SDE and ODE Background

For notation references, this subsection briefly recaps the SDE and ODE formulation of diffusion models by Song et al. (2021b).

The forward-time SDE of the diffusion process can be derived by taking an infinitesimal step size $\Delta t$ when applying the transition Gaussian kernel in Eq. (1), which yields:

$$\mathrm{d}\boldsymbol{x} = f_t \boldsymbol{x}_t \, \mathrm{d}t + g_t \, \mathrm{d}\boldsymbol{w}_t \qquad (22)$$

where $f_t = \frac{1}{\alpha_t}\frac{\mathrm{d}\alpha_t}{\mathrm{d}t}$, $g_t = \sqrt{\frac{\mathrm{d}\beta_{t,\Delta t}}{\mathrm{d}\Delta t}}\big|_{\Delta t=0}$, and $\boldsymbol{w}_t$ is the standard Wiener process. The corresponding reverse-time SDE is:

$$\mathrm{d}\boldsymbol{x}_t = \left(f_t \boldsymbol{x}_t - g_t^2 \boldsymbol{s}_t(\boldsymbol{x}_t)\right) \mathrm{d}t + g_t \, \mathrm{d}\bar{\boldsymbol{w}}_t, \qquad (23)$$

with the score function $\boldsymbol{s}_t(\boldsymbol{x}_t) := \nabla_{\boldsymbol{x}_t} \log p(\boldsymbol{x}_t) = \mathbb{E}_{\boldsymbol{x}_0 \sim p(\boldsymbol{x}_0|\boldsymbol{x}_t)}\left[-\frac{\alpha_t \boldsymbol{x}_0 - \boldsymbol{x}_t}{\sigma_t^2}\right]$ and the reverse-time standard Wiener process $\bar{\boldsymbol{w}}$. Using the Fokker–Planck equation, we can prove that the time evolution of the PDF $p(\boldsymbol{x}_t)$ described by the SDEs is equivalent to that described by an ODE:

$$\mathrm{d}\boldsymbol{x}_t = \left(f_t \boldsymbol{x}_t - \frac{1}{2} g_t^2 \boldsymbol{s}_t(\boldsymbol{x}_t)\right) \mathrm{d}t, \qquad (24)$$

which maps the noise $\boldsymbol{x}_t$ to a deterministic data point $\boldsymbol{x}_0$.

### C.2. Proof of Theorem 3.1

*Proof.* Let $p(\boldsymbol{u})$ be the PDF of an arbitrary distribution on $\mathbb{R}^D$, and $\{\boldsymbol{\Sigma}_k\}_{k=1}^K$ be a set of arbitrary $D \times D$ symmetric

positive definite matrices. We aim to show that if

$$\{a_k^*, \boldsymbol{\mu}_k^*\} = \arg\min_{\{a_k, \boldsymbol{\mu}_k\}} \mathbb{E}_{\boldsymbol{u} \sim p(\boldsymbol{u})}\left[-\log \sum_{k=1}^K A_k \mathcal{N}(\boldsymbol{u}; \boldsymbol{\mu}_k, \boldsymbol{\Sigma}_k)\right],$$
$$(25)$$

with $a_k \in \mathbb{R}$, $\boldsymbol{\mu}_k \in \mathbb{R}^D$, $A_k = \frac{\exp a_k}{\sum_{k=1}^K \exp a_k}$, then the mean alignment property holds:

$$\sum_{k=1}^K A_k^* \boldsymbol{\mu}_k^* = \mathbb{E}_{\boldsymbol{u} \sim p(\boldsymbol{u})}[\boldsymbol{u}], \qquad (26)$$

with $A_k^* = \frac{\exp a_k^*}{\sum_{k=1}^K \exp a_k^*}$.

For brevity, we define

$$q_k(\boldsymbol{u}) := A_k^* \mathcal{N}(\boldsymbol{u}; \boldsymbol{\mu}_k^*, \boldsymbol{\Sigma}_k), \qquad (27)$$

$$q(\boldsymbol{u}) := \sum_{k=1}^K q_k(\boldsymbol{u}). \qquad (28)$$

Since $\{a_k^*, \boldsymbol{\mu}_k^*\}$ minimizes the objective, the first-order optimality conditions imply

$$\frac{\partial \mathbb{E}_{\boldsymbol{u} \sim p(\boldsymbol{u})}[-\log q(\boldsymbol{u})]}{\partial a_k^*}$$
$$= \mathbb{E}_{\boldsymbol{u} \sim p(\boldsymbol{u})}\left[\frac{\partial - \log q(\boldsymbol{u})}{\partial a_k^*}\right]$$
$$= \mathbb{E}_{\boldsymbol{u} \sim p(\boldsymbol{u})}\left[A_k^* - \frac{q_k(\boldsymbol{u})}{q(\boldsymbol{u})}\right]$$
$$= A_k^* - \mathbb{E}_{\boldsymbol{u} \sim p(\boldsymbol{u})}\left[\frac{q_k(\boldsymbol{u})}{q(\boldsymbol{u})}\right]$$
$$= 0, \qquad (29)$$

and

$$\frac{\partial \mathbb{E}_{\boldsymbol{u} \sim p(\boldsymbol{u})}[-\log q(\boldsymbol{u})]}{\partial \boldsymbol{\mu}_k^*}$$
$$= \mathbb{E}_{\boldsymbol{u} \sim p(\boldsymbol{u})}\left[\frac{\partial - \log q(\boldsymbol{u})}{\partial \boldsymbol{\mu}_k^*}\right]$$
$$= \mathbb{E}_{\boldsymbol{u} \sim p(\boldsymbol{u})}\left[\frac{q_k(\boldsymbol{u})}{q(\boldsymbol{u})}\boldsymbol{\Sigma}_k^{-\frac{1}{2}}(\boldsymbol{\mu}_k^* - \boldsymbol{u})\right]$$
$$= \boldsymbol{\Sigma}_k^{-\frac{1}{2}}\left(\boldsymbol{\mu}_k^* \mathbb{E}_{\boldsymbol{u} \sim p(\boldsymbol{u})}\left[\frac{q_k(\boldsymbol{u})}{q(\boldsymbol{u})}\right] - \mathbb{E}_{\boldsymbol{u} \sim p(\boldsymbol{u})}\left[\frac{q_k(\boldsymbol{u})}{q(\boldsymbol{u})}\boldsymbol{u}\right]\right)$$
$$= \boldsymbol{0}. \qquad (30)$$

From the above, it follows that

$$A_k^* = \mathbb{E}_{\boldsymbol{u} \sim p(\boldsymbol{u})}\left[\frac{q_k(\boldsymbol{u})}{q(\boldsymbol{u})}\right], \qquad (31)$$

$$\boldsymbol{\mu}_k^* \mathbb{E}_{\boldsymbol{u} \sim p(\boldsymbol{u})}\left[\frac{q_k(\boldsymbol{u})}{q(\boldsymbol{u})}\right] = \mathbb{E}_{\boldsymbol{u} \sim p(\boldsymbol{u})}\left[\frac{q_k(\boldsymbol{u})}{q(\boldsymbol{u})}\boldsymbol{u}\right]. \qquad (32)$$

Substituting Eq. (31) into Eq. (32), we have

$$A_k^* \boldsymbol{\mu}_k^* = \mathbb{E}_{\boldsymbol{u} \sim p(\boldsymbol{u})} \left[ \frac{q_k(\boldsymbol{u})}{q(\boldsymbol{u})} \boldsymbol{u} \right]. \tag{33}$$

Summing over all $k$, we conclude that

$$\sum_{k=1}^{K} A_k^* \boldsymbol{\mu}_k^* = \mathbb{E}_{\boldsymbol{u} \sim p(\boldsymbol{u})} \left[ \frac{\sum_{k=1}^{K} q_k(\boldsymbol{u})}{q(\boldsymbol{u})} \boldsymbol{u} \right] = \mathbb{E}_{\boldsymbol{u} \sim p(\boldsymbol{u})}[\boldsymbol{u}]. \tag{34}$$

$\square$

### C.3. Derivation of the Reverse Transition Distribution

Given the ground truth data distribution $p(\boldsymbol{x}_0)$ and the forward diffusion Gaussian $p(\boldsymbol{x}_t|\boldsymbol{x}_0) = \mathcal{N}(\boldsymbol{x}_t; \alpha_t \boldsymbol{x}_0, \sigma_t^2 \boldsymbol{I})$, the ground truth denoising distribution $p(\boldsymbol{x}_0|\boldsymbol{x}_t)$ can be derived using Bayes' theorem:

$$p(\boldsymbol{x}_0|\boldsymbol{x}_t) = \frac{p(\boldsymbol{x}_0)p(\boldsymbol{x}_t|\boldsymbol{x}_0)}{p(\boldsymbol{x}_t)}. \tag{35}$$

Conversely, let us assume that we know the ground truth denoising distribution $p(\boldsymbol{x}_0|\boldsymbol{x}_t)$. By rearranging Eq. (35), we can derive the data distribution as:

$$p(\boldsymbol{x}_0) = \frac{p(\boldsymbol{x}_t)p(\boldsymbol{x}_0|\boldsymbol{x}_t)}{p(\boldsymbol{x}_t|\boldsymbol{x}_0)}. \tag{36}$$

With the data distribution, we can apply the forward diffusion process and derive the noisy data distribution at $t - \Delta t$:

$$p(\boldsymbol{x}_{t-\Delta t}) = \int_{\mathbb{R}^D} p(\boldsymbol{x}_{t-\Delta t}|\boldsymbol{x}_0)p(\boldsymbol{x}_0) \, \mathrm{d}\boldsymbol{x}_0$$

$$= \int_{\mathbb{R}^D} p(\boldsymbol{x}_{t-\Delta t}|\boldsymbol{x}_0) \frac{p(\boldsymbol{x}_t)p(\boldsymbol{x}_0|\boldsymbol{x}_t)}{p(\boldsymbol{x}_t|\boldsymbol{x}_0)} \, \mathrm{d}\boldsymbol{x}_0. \tag{37}$$

Finally, the reverse transition distribution $p(\boldsymbol{x}_{t-\Delta t}|\boldsymbol{x}_t)$ can be derived using Bayes' theorem:

$$p(\boldsymbol{x}_{t-\Delta t}|\boldsymbol{x}_t)$$
$$= \frac{p(\boldsymbol{x}_{t-\Delta t})p(\boldsymbol{x}_t|\boldsymbol{x}_{t-\Delta t})}{p(\boldsymbol{x}_t)}$$
$$= \int_{\mathbb{R}^D} \frac{p(\boldsymbol{x}_t|\boldsymbol{x}_{t-\Delta t})p(\boldsymbol{x}_{t-\Delta t}|\boldsymbol{x}_0)}{p(\boldsymbol{x}_t|\boldsymbol{x}_0)} p(\boldsymbol{x}_0|\boldsymbol{x}_t) \, \mathrm{d}\boldsymbol{x}_0, \tag{38}$$

where $p(\boldsymbol{x}_t|\boldsymbol{x}_{t-\Delta t})$ is the forward transition Gaussian defined in Eq. (1). The term $\frac{p(\boldsymbol{x}_t|\boldsymbol{x}_{t-\Delta t})p(\boldsymbol{x}_{t-\Delta t}|\boldsymbol{x}_0)}{p(\boldsymbol{x}_t|\boldsymbol{x}_0)}$ can be fused into one Gaussian PDF using the conflation operation described in § D.1, which yields:

$$p(\boldsymbol{x}_{t-\Delta t}|\boldsymbol{x}_t, \boldsymbol{x}_0) = \mathcal{N}(\boldsymbol{x}_{t-\Delta t}; c_1 \boldsymbol{x}_t + c_2 \boldsymbol{x}_0, c_3 \boldsymbol{I}), \tag{39}$$

with the coefficients $c_1, c_2, c_3$ defined in § 3.3. Therefore, Eq. (38) can be simplified into:

$$p(\boldsymbol{x}_{t-\Delta t}|\boldsymbol{x}_t) = \int_{\mathbb{R}^D} p(\boldsymbol{x}_{t-\Delta t}|\boldsymbol{x}_t, \boldsymbol{x}_0)p(\boldsymbol{x}_0|\boldsymbol{x}_t) \, \mathrm{d}\boldsymbol{x}_0, \tag{40}$$

which is a convolution between $p(\boldsymbol{x}_{t-\Delta t}|\boldsymbol{x}_t, \boldsymbol{x}_0)$ and $p(\boldsymbol{x}_0|\boldsymbol{x}_t)$. Sampling from $p(\boldsymbol{x}_{t-\Delta t}|\boldsymbol{x}_t)$ can therefore be simulated by first sampling $\boldsymbol{x}_0 \sim p(\boldsymbol{x}_0|\boldsymbol{x}_t)$ and then sampling $\boldsymbol{x}_{t-\Delta t} \sim p(\boldsymbol{x}_{t-\Delta t}|\boldsymbol{x}_t, \boldsymbol{x}_0)$.

In general, given any empirical denoising distribution $q_{\boldsymbol{\theta}}(\boldsymbol{x}_0|\boldsymbol{x}_t)$, we can perform stochastic denoising sampling by substituting $p(\boldsymbol{x}_0|\boldsymbol{x}_t) \approx q_{\boldsymbol{\theta}}(\boldsymbol{x}_0|\boldsymbol{x}_t)$ into the above sampling process. Specifically for GMFlow, $q_{\boldsymbol{\theta}}(\boldsymbol{x}_0|\boldsymbol{x}_t)$ is a GM PDF. § D.3 shows that the convolution of a GM and a Gaussian is also a GM with analytically derived parameters. Therefore, GMFlow models the reverse transition distribution as a GM $q_{\boldsymbol{\theta}}(\boldsymbol{x}_{t-\Delta t}|\boldsymbol{x}_t)$, given by Eq. (9).

### C.4. Additional Analysis of the Reverse Transition GM

By computing the first-order Taylor approximation of Eq. (9) w.r.t. $\Delta t$, we can re-write the GM reverse transition distribution as:

$$q_{\boldsymbol{\theta}}(\boldsymbol{x}_{t-\Delta t}|\boldsymbol{x}_t)$$
$$= \sum_{k=1}^{K} A_k \mathcal{N}\big(\boldsymbol{x}_{t-\Delta t}; \boldsymbol{x}_t - (f_t \boldsymbol{x}_t - g_t^2 \boldsymbol{\mu}_{\mathrm{s}k})\Delta t, g_t^2 \Delta t \boldsymbol{I}\big)$$
$$+ O(\Delta t)$$
$$= \mathcal{N}\left(\boldsymbol{x}_{t-\Delta t}; \boldsymbol{x}_t - \left(f_t \boldsymbol{x}_t - g_t^2 \sum_{k=1}^{K} A_k \boldsymbol{\mu}_{\mathrm{s}k}\right)\Delta t, g_t^2 \Delta t \boldsymbol{I}\right)$$
$$+ O(\Delta t), \tag{41}$$

where $\boldsymbol{\mu}_{\mathrm{s}k} := -\frac{1}{\sigma_t}(\boldsymbol{x}_t + \alpha_t \boldsymbol{\mu}_k)$. The term $\sum_{k=1}^{K} A_k \boldsymbol{\mu}_{\mathrm{s}k}$ represents the model's prediction of the score function $\boldsymbol{s}_t(\boldsymbol{x}_t)$, and is a linear transformation of the predicted mean velocity $\sum_{k=1}^{K} A_k \boldsymbol{\mu}_k$. Recursively sampling $x_{t-\Delta t}$ using the first-order approximation in Eq. (41) is equivalent to solving the reverse-time SDE in Eq. (23) (with predicted score) using the Euler–Maruyama method, a simple first-order SDE solver.

Eq. (41) is consistent with the standard diffusion model assumption that the reverse transition distribution is approximately Gaussian for small $\Delta t$. Even with GM parameterization, SDE sampling is primarily influenced by the mean prediction when $\Delta t$ is sufficiently small. This underscores the significance of mean alignment not only for ODE solving but also for SDE solving.

### C.5. Derivation of $\hat{q}(\boldsymbol{x}_0|\boldsymbol{x}_\tau)$

Let us assume that we know the ground truth denoising distribution at $\boldsymbol{x}_t$, i.e., $p(\boldsymbol{x}_0|\boldsymbol{x}_t)$. Eq. (36) shows us how to derive the data distribution $p(\boldsymbol{x}_0)$ from $p(\boldsymbol{x}_0|\boldsymbol{x}_t)$. From $p(\boldsymbol{x}_0)$, the denoising distribution at $\boldsymbol{x}_\tau$ can be derived using

Bayes' theorem:

$$
\begin{aligned}
p(\boldsymbol{x}_0|\boldsymbol{x}_\tau) &= \frac{p(\boldsymbol{x}_0)p(\boldsymbol{x}_\tau|\boldsymbol{x}_0)}{p(\boldsymbol{x}_\tau)} \\
&= \frac{p(\boldsymbol{x}_t)p(\boldsymbol{x}_0|\boldsymbol{x}_t)p(\boldsymbol{x}_\tau|\boldsymbol{x}_0)}{p(\boldsymbol{x}_t|\boldsymbol{x}_0)p(\boldsymbol{x}_\tau)} \\
&= \frac{p(\boldsymbol{x}_\tau|\boldsymbol{x}_0)}{Z \cdot p(\boldsymbol{x}_t|\boldsymbol{x}_0)}p(\boldsymbol{x}_0|\boldsymbol{x}_t),
\end{aligned}
\tag{42}
$$

where $Z = \frac{p(\boldsymbol{x}_\tau)}{p(\boldsymbol{x}_t)}$ is a normalization factor. Substituting $p(\boldsymbol{x}_0|\boldsymbol{x}_t) \approx q_{\boldsymbol{\theta}}(\boldsymbol{x}_0|\boldsymbol{x}_t)$ into the above equation yields the denoising distribution conversion rule in Eq. (10). The conversion involves conflations of Gaussians and a GM, which can be approached analytically as shown in § D.

# D. Gaussian Mixture Math References

## D.1. Conflation of Two Gaussians

Let $p_1(\boldsymbol{x}) = \mathcal{N}(\boldsymbol{x}; \boldsymbol{\mu}_1, \boldsymbol{\Sigma}_1)$, $p_2(\boldsymbol{x}) = \mathcal{N}(\boldsymbol{x}; \boldsymbol{\mu}_2, \boldsymbol{\Sigma}_2)$ be two multivariate Gaussian PDFs, their powered conflation is defined as:

$$
p'(\boldsymbol{x}) = \frac{p_1^{\gamma_1}(\boldsymbol{x})p_2^{\gamma_2}(\boldsymbol{x})}{Z},
\tag{43}
$$

where $\gamma_1, \gamma_2 \in \mathbb{R}$, assuming $\gamma_1 \boldsymbol{\Sigma}_1^{-1} + \gamma_2 \boldsymbol{\Sigma}_2^{-1}$ is positive definite, and $Z = \int_{\mathbb{R}^D} p_1^{\gamma_1}(\boldsymbol{x})p_2^{\gamma_2}(\boldsymbol{x}) \, d\boldsymbol{x}$ is a normalization factor. It's easy to prove that, $p'(\boldsymbol{x})$ can also be expressed as a Gaussian $\mathcal{N}(\boldsymbol{x}; \boldsymbol{\mu}', \boldsymbol{\Sigma}')$, with the new parameters:

$$
\boldsymbol{\Sigma}' = \left(\gamma_1 \boldsymbol{\Sigma}_1^{-1} + \gamma_2 \boldsymbol{\Sigma}_2^{-1}\right)^{-1},
\tag{44}
$$

$$
\boldsymbol{\mu}' = \boldsymbol{\Sigma}'(\gamma_1 \boldsymbol{\Sigma}_1^{-1}\boldsymbol{\mu}_1 + \gamma_2 \boldsymbol{\Sigma}_2^{-1}\boldsymbol{\mu}_2).
\tag{45}
$$

## D.2. Conflation of a Gaussian and a GM

Let $p_1(\boldsymbol{x}) = \mathcal{N}(\boldsymbol{x}; \boldsymbol{\mu}, \boldsymbol{\Sigma})$ be a Gaussian PDF, and $p_2(\boldsymbol{x}) = \sum_{k=1}^K A_k \mathcal{N}(\boldsymbol{x}; \boldsymbol{\mu}_k, \boldsymbol{\Sigma}_k)$ be a GM PDF, where $A_k = \frac{\exp a_k}{\sum_{k=1}^K \exp a_k}$ with logit $a_k$. Their conflation is defined as:

$$
p'(\boldsymbol{x}) = \frac{p_1(\boldsymbol{x})p_2(\boldsymbol{x})}{Z},
\tag{46}
$$

where $Z = \int_{\mathbb{R}^D} p_1(\boldsymbol{x})p_2(\boldsymbol{x}) \, d\boldsymbol{x}$ is a normalization factor. Since the GM PDF is a sum of Gaussians, the conflation of a Gaussian and a GM expands to a sum of conflations of Gaussians, which simplifies to a sum of Gaussians. Therefore, $p'(\boldsymbol{x})$ can also be expressed as a GM $\sum_{k=1}^K A_k' \mathcal{N}(\boldsymbol{x}; \boldsymbol{\mu}_k', \boldsymbol{\Sigma}_k')$, with the new parameters:

$$
\boldsymbol{\Sigma}_k' = (\boldsymbol{\Sigma}^{-1} + \boldsymbol{\Sigma}_k^{-1})^{-1},
\tag{47}
$$

$$
\boldsymbol{\mu}_k' = \boldsymbol{\Sigma}_k'(\boldsymbol{\Sigma}^{-1}\boldsymbol{\mu} + \boldsymbol{\Sigma}_k^{-1}\boldsymbol{\mu}_k),
\tag{48}
$$

$$
A_k' = \frac{\exp a_k'}{\sum_{k=1}^K \exp a_k'},
\tag{49}
$$

where the new logit $a_k'$ is given by:

$$
a_k' = a_k - \frac{1}{2}(\boldsymbol{\mu} - \boldsymbol{\mu}_k)^{\mathrm{T}}(\boldsymbol{\Sigma} + \boldsymbol{\Sigma}_k)^{-1}(\boldsymbol{\mu} - \boldsymbol{\mu}_k).
\tag{50}
$$

## D.3. Convolution of a Gaussian and a GM

Let $p(\boldsymbol{x}_1|\boldsymbol{x}_2) = \mathcal{N}(\boldsymbol{x}_1; \boldsymbol{\mu} + c\boldsymbol{x}_2, \boldsymbol{\Sigma})$ be a conditional Gaussian PDF, where $c \in \mathbb{R}$ is a linear coefficient, and $p(\boldsymbol{x}_2) = \sum_{k=1}^K A_k \mathcal{N}(\boldsymbol{x}_2; \boldsymbol{\mu}_k, \boldsymbol{\Sigma}_k)$ be a GM PDF, where $A_k = \frac{\exp a_k}{\sum_{k=1}^K \exp a_k}$ with logit $a_k$. Their convolution yields the marginal PDF of $\boldsymbol{x}_1$:

$$
p(\boldsymbol{x}_1) = \int_{\mathbb{R}^D} p(\boldsymbol{x}_1|\boldsymbol{x}_2)p(\boldsymbol{x}_2) \, d\boldsymbol{x}_2.
\tag{51}
$$

Since the GM PDF is a sum of Gaussians, the convolution of a Gaussian and a GM expands to a sum of convolution of Gaussians, which simplifies to a sum of Gaussians. Therefore $p(\boldsymbol{x}_1)$ can also be expressed as a GM $\sum_{k=1}^K A_k \mathcal{N}(\boldsymbol{x}; \boldsymbol{\mu}_k', \boldsymbol{\Sigma}_k')$, with the new parameters:

$$
\boldsymbol{\Sigma}_k' = \boldsymbol{\Sigma} + c^2 \boldsymbol{\Sigma}_k,
\tag{52}
$$

$$
\boldsymbol{\mu}_k' = \boldsymbol{\mu} + c\boldsymbol{\mu}_k.
\tag{53}
$$

# E. Notation

*Table 6.* A summary of frequently used notations.

| Notation | | Description |
|---|---|---|
| $D$ | | Data dimension. |
| $t$ | $\in [0, T]$ | Diffusion time. Flow matching models define $T \coloneqq 1$. |
| $\alpha_t$ | | Noise schedule coefficient. Flow matching models define $\alpha_t \coloneqq 1 - t$. |
| $\sigma_t$ | | Noise schedule coefficient. Flow matching models define $\sigma_t \coloneqq t$. |
| $\boldsymbol{\epsilon}$ | | Standard Gaussian noise. |
| $\boldsymbol{x}, \boldsymbol{x}_0$ | $\in \mathbb{R}^D$ | Data. |
| $\boldsymbol{x}_t$ | $\coloneqq \alpha_t \boldsymbol{x} + \sigma_t \boldsymbol{\epsilon}$ | Noisy data. |
| $p(\boldsymbol{x}), p(\boldsymbol{x}_0)$ | | PDF of data (ground truth). |
| $p(\boldsymbol{x}_t)$ | | PDF of noisy data (ground truth). |
| $p(\boldsymbol{x}_t\|\boldsymbol{x}_0)$ | $= \mathcal{N}(\boldsymbol{x}_t; \alpha_t \boldsymbol{x}_0, \sigma_t^2 \boldsymbol{I})$ | PDF of forward diffusion distribution. |
| $p(\boldsymbol{x}_t\|\boldsymbol{x}_{t-\Delta t})$ | $= \mathcal{N}(\boldsymbol{x}_t; \frac{\alpha_t}{\alpha_{t-\Delta t}} \boldsymbol{x}_0, \beta_{t,\Delta t} \boldsymbol{I})$ | PDF of forward transition distribution. |
| $\beta_{t,\Delta t}$ | $= \sigma_t^2 - \frac{\alpha_t^2}{\alpha_{t-\Delta t}^2} \sigma_{t-\Delta t}^2$ | Variance of forward transition distribution. |
| $p(\boldsymbol{x}_0\|\boldsymbol{x}_t)$ | $= \frac{p(\boldsymbol{x}_0)p(\boldsymbol{x}_t\|\boldsymbol{x}_0)}{p(\boldsymbol{x}_t)}$ | PDF of reverse denoising distribution (ground truth). |
| $p(\boldsymbol{x}_{t-\Delta t}\|\boldsymbol{x}_t)$ | $= \frac{p(\boldsymbol{x}_{t-\Delta t})p(\boldsymbol{x}_t\|\boldsymbol{x}_{t-\Delta t})}{p(\boldsymbol{x}_t)}$ | PDF of reverse transition distribution (ground truth). |
| $\boldsymbol{\theta}$ | | Neural network parameters. |
| $q_{\boldsymbol{\theta}}(\boldsymbol{x}_0\|\boldsymbol{x}_t)$ | | PDF of reverse denoising distribution, predicted by a network. |
| $q_{\boldsymbol{\theta}}(\boldsymbol{x}_{t-\Delta t}\|\boldsymbol{x}_t)$ | | PDF of reverse transition distribution, predicted by a network. |
| $\boldsymbol{u}$ | $\coloneqq \frac{\boldsymbol{x}_t - \boldsymbol{x}_0}{\sigma_t}$ | Random flow velocity. |
| $\mathbb{E}_{\boldsymbol{x}_0 \sim p(\boldsymbol{x}_0\|\boldsymbol{x}_t)}[\boldsymbol{u}]$ | | Mean flow velocity at $\boldsymbol{x}_t$ (ground truth). |
| $p(\boldsymbol{u}\|\boldsymbol{x}_t)$ | | PDF of velocity distribution at $\boldsymbol{x}_t$ (ground truth), derived from $p(\boldsymbol{x}_0\|\boldsymbol{x}_t)$. |
| $q_{\boldsymbol{\theta}}(\boldsymbol{u}\|\boldsymbol{x}_t)$ | | PDF of velocity distribution at $\boldsymbol{x}_t$, predicted by a network. |
| $\boldsymbol{\mu}_{\boldsymbol{\theta}}(\boldsymbol{x}_t)$ | | Mean flow velocity at $\boldsymbol{x}_t$, predicted by a network. |
| $K$ | | Number of Gaussian components. |
| $k$ | | Index of Gaussian components. |
| $A_k$ | $\coloneqq \frac{\exp a_k}{\sum_{k=1}^K \exp a_k}$. | Mixture weight of the $k$-th Gaussian component. |
| $\boldsymbol{\mu}_k$ | | Mean of the $k$-th Gaussian component. |
| $\boldsymbol{\Sigma}_k$ | | Covariance of the $k$-th Gaussian component. GMFlow defines $\boldsymbol{\Sigma}_k \coloneqq s^2 \boldsymbol{I}$. |
| $a_k$ | | Pre-activation logit. |
| $s$ | | Shared standard deviation. |
| $\boldsymbol{\mu}_{\mathrm{x}k}$ | $\coloneqq \boldsymbol{x}_t - \sigma_t \boldsymbol{\mu}_k$ | Mean of the $k$-th Gaussian component after $\boldsymbol{u}$-to-$\boldsymbol{x}_0$ conversion. |
| $s_{\mathrm{x}}$ | $\coloneqq \sigma_t s$ | Shared standard deviation after $\boldsymbol{u}$-to-$\boldsymbol{x}_0$ conversion. |
| $\boldsymbol{c}$ | | Condition. |
| $w$ | $\in [1, +\infty)$ | CFG scale. |
| $\tilde{w}$ | $\in [0, 1)$ | Probabilistic guidance scale. |
| $\hat{\boldsymbol{x}}$ | | Intermediate sample of denoised data. |
| $NFE$ | | Number of function (network) evaluations, a.k.a. sampling steps. |
| $n$ | | Number of sub-steps in GM-ODE solvers. |
| $\tau$ | | Sub-step diffusion time in GM-ODE solvers. |
| $\hat{q}(\boldsymbol{x}_0\|\boldsymbol{x}_\tau)$ | | PDF of reverse denoising distribution, derived from $q_{\boldsymbol{\theta}}(\boldsymbol{x}_0\|\boldsymbol{x}_t)$ (Eq. (10)). |
| $\hat{q}(\boldsymbol{x}_0\|\boldsymbol{x}_t)$ | | PDF of reverse denoising distribution, derived from $q_{\boldsymbol{\theta}}(\boldsymbol{x}_0\|\boldsymbol{x}_{t+\Delta t})$. |
| $\lambda$ | $\in (0, 1]$ | Transition ratio. |

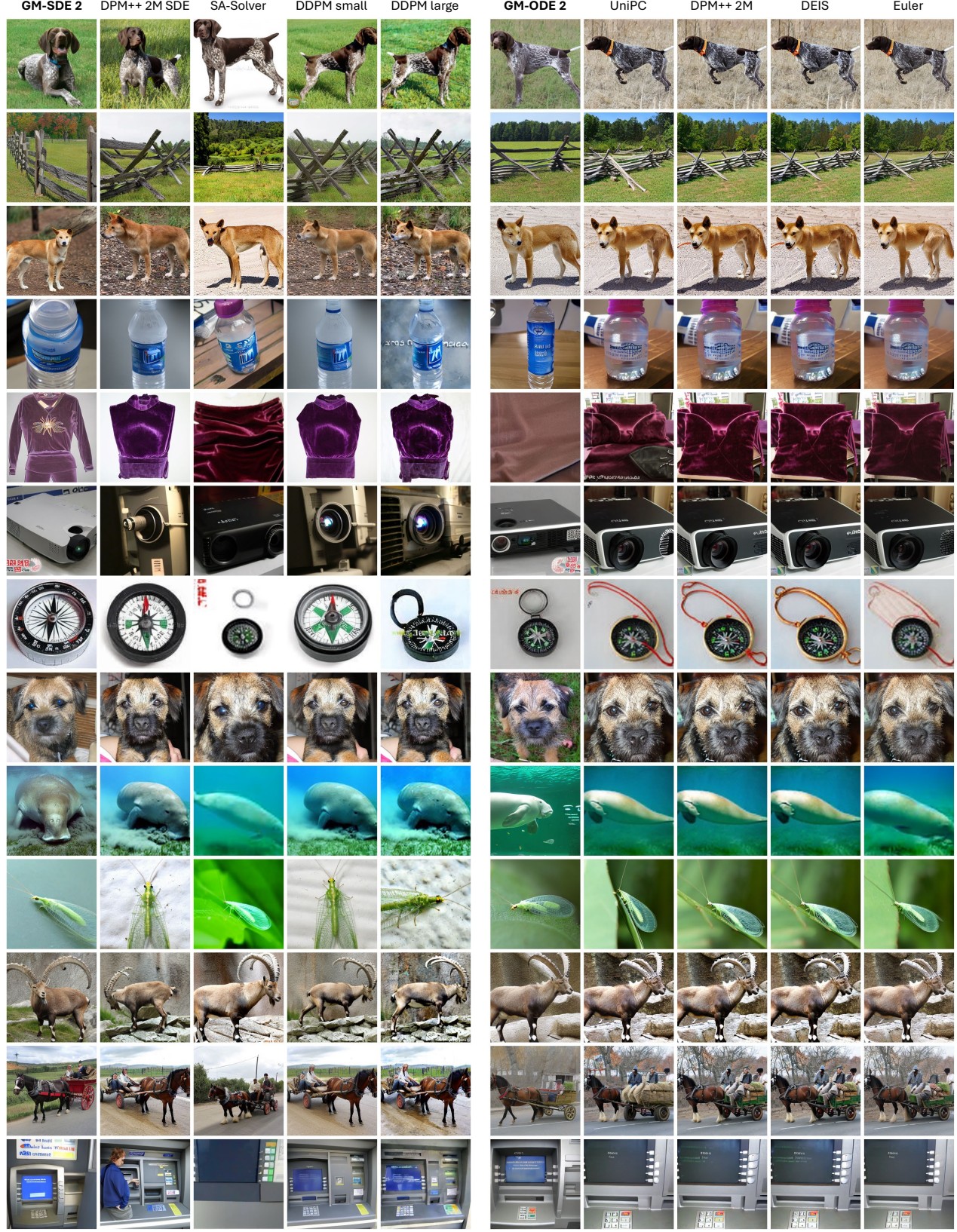

*Figure 11.* Uncurated samples (at best Precision, $NFE = 32$) from GMFlow and vanilla flow matching baselines using different solvers. The images generated by GMFlow exhibit more natural color saturation and, in some cases, improved structural coherence.

