# OpenReview forum: "Gaussian Mixture Flow Matching Models"
_ICML.cc/2025/Conference — ICML 2025 poster_

### Official Review · Reviewer_A5dP · 2025-03-05

**Overall Recommendation:** 4

**Summary:**

This paper introduces GM-Flow, a novel variant of flow matching that explicitly parameterizes the entire velocity distribution using a mixture of Gaussians, rather than learning only the mean velocity as in conventional approaches. Unlike CFG, which extrapolates class-conditional and unconditional velocities, GM-Flow employs GM morphing to obtain the class-guided velocity, effectively avoiding extreme cases that lead to oversaturation in generated images. Additionally, the paper proposes a GM solver for sampling. The method has been validated through large-scale experiments on ImageNet, demonstrating strong scalability.

**Claims And Evidence:**

Yes

**Essential References Not Discussed:**

No.

**Experimental Designs Or Analyses:**

Yes.

**Methods And Evaluation Criteria:**

Yes

**Other Comments Or Suggestions:**

Please see the section of [Other Strengths and Weaknesses].

**Other Strengths And Weaknesses:**

## Strengths
1. The GM morphing approach is quite novel.
2. The performance of the method is quite impressive and the method seems to scale well to large-scale datasets and models.

## Weaknesses and Questions

1. This paper lacks a direct experimental comparison with GMS (Guo et al. 2023).
2. When using GM-SDE and perform spectral sampling, it seems to entail inverse KR transport. If I understand correctly, this step needs to be solved numerically, am I correct? In this case, the numerical steps should also be considered as a part of the computational complexity, instead of simply use NFE as the complexity metric. Please provide more detailed explanation on how the spectral sampling is performed and how complex it is.

Most of my weaknesses/questions come from the Section 3.2. This section needs to be expanded, or a section in Appendix should be added to further elaborate the content in Section 3.2 in details.

3. Why constructing a Gaussian mask using two surrogate Gaussians instead of the full GMs? Is it due to computational complexity when utilizing the full GMs?

4. In Equation (7), how to interpret $g(u)$ in the context of "class-conditional guidance"? I mean, CFG basically add a shift $w(\mu_c - \mu_u)$ to the unconditional "basis vector" $\mu_u$, right? But in Equation (7), the "basis vector" is $\mu_c$, and the shift is $\tilde{w} (\mu_c - \mu_u) / ||\mu_c - \mu_u|| \sqrt{D}$. What is the intuition behind this design of guided Gaussian?

5. In Equation (7), why do you scale the shift $\Delta \mu_n$ by $s_c$?

6. In Equation (7), how do you obtain $\mu_c$ and $\mu_u$ from the model? Do they obtained by minimizing KL divergence between the surrogate and the learned GM $q_\theta(u|x_t)$?

6. You said you use the orthogonal projection trick to obtain better sampling quality, is this trick applied to Equation (7)? Do you apply it on $\Delta \mu_n$?

8. In Equation (8), why do they have $ \mathcal{N}(\mathbf{u}; \mathbf{\mu}_c, s_c^2 I) $ in the denominator? The Appendix C only illustrate how to compute the $\frac{g(u) q _{\theta}(u|x _t, c)}{Z}$. If there is a Gaussian PDF in the denominator, will Equation (8) still yield a GM, and still analytically computable? My thought is that $ g(u) / \mathcal{N}(u; \mu_c, s_c^2 I) $ is proportional to another gaussian, so in the end it is a gaussian multiplying with a GM, thereby producing another GM. Am I correct? **Please provide the analytical solution of Equation (8).**

**Questions For Authors:**

1. **Please upload the code during the rebuttal.**

2. In Eq (6), what is $\mathbf{x}$, shouldn't it be $\mathbf{u}$? and what is $D$, the dimension of $\mathbf{x}$? In previous sections you use $d$.

3. What do "DDPM small" and "DDPM large" mean in Figure 4 (a)? The ancestral sampling of DDPM with different standard deviations $\sigma_t = \beta_t$ or $\tilde{\beta}_t $ of the isotropic Gaussian? Please explain how you apply DDPM sampling scheme to flow models in details.

4. Is $\mathcal{L}_{\text{spec}}$ used for training even if GM-ODE will be used for sampling? How are the two losses  $\mathcal{ L } _ {\text{trans }}$ and $\mathcal{L} _ {\text{spec}}$ weighted?

5. In the paragraph "$u$-to-$x_0$ reparametrization", in Line 192-193, the second parameter of the GM component should be $s_x^2$. The same for the $q_\theta(x_0|x_t)$ on the right hand side of the same page, in the paragraph "GM-SDE solver".

6. In Appendix A.4, is there any typo in the definition of $\mathbf{\mu}_{s_k}$? Please double check it.

7. In Figure 3, is $s$ shared across all spatial locations? If this is the case, then what does "the mean of per-pixel GM variances" refer to? Additionally, please explain the modelling process of $\mathbf{s}_{\text{F}}$ more detailed.

**Relation To Broader Scientific Literature:**

This method extends conventional flow matching and addresses a key limitation by explicitly parameterizing the velocity distribution, offering a more robust approach to handling class-guided velocity.

**Theoretical Claims:**

See Weaknesses

---

> ### Author Rebuttal · Authors · 2025-04-01
>
> Thank you for your thoughtful review. We have uploaded a **revised manuscript** and essential **code** in this anonymous link (full code will be released upon publication):
>
> https://anonymous.4open.science/r/anonymous_gmflow-63FE
> backup: https://limewire.com/d/CgAn9#jkBxDmC3qh
>
> ### **Weaknesses and Questions**
>
> > 1. Comparison with GMS.
>
> We’ve made a direct comparison with GMS and reported the results on unconditional CIFAR-10 image generation in Appendix B.3 of the revised manuscript. We train GMFlow with $K=2$ from scratch using the same U-Net backbone as GMS. We choose CIFAR-10 because re-training an ImageNet model using the backbone of GMS is expensive and time-consuming.
>
> Here are the FID results. GMFlow significantly outperforms GMS and other moment-matching methods in few-step sampling.  The competitor results are from the original GMS paper.
>
> |**NFE**| 10 | 25 | 50 | 100 |
> |-|-|-|-|-|
> | DDPM large | 205.31 | 84.71 | 37.35 | 14.81 |
> | DDPM small | 34.76 | 16.18 | 11.11 | 8.38 |
> | SN-DDPM | 16.33 | 6.05 | 4.19  | 3.83 |
> | GMS | 13.80  | 5.48 | 4.00 | 3.46 |
> | **GM-SDE 2 (ours)**| 9.11 | 4.16 | 3.79 | 3.76 |
>
> > 2. Computational complexity of KR transport
>
> We would like to point out that KR transport is highly efficient since it's basically 1D CDF mappings. In the paper, we stated that GMFlow incurs only 0.005 sec of overhead per step, which is minimal compared to the total inference time of 0.39 sec per step (most of which is spent on DiT). This includes the KR transport. The computation complexity of per-pixel KR transport is $O(K\cdot C^2)$ ($C$ is the channel size). We have added more details on spectral sampling in Figure 9 of the revised manuscript.
>
> > Section 3.2 needs to be expanded
>
> Thank you for the suggestion. In the revised manuscript, we have fully rewritten and expanded Section 3.2 to make it more clear. We kindly refer the reviewer to the revised manuscript for more details.
>
> > 3. Why construct a Gaussian mask using two surrogate Gaussians?
>
> The reasons are twofold:
> - The division of a GM PDF by another GM PDF doesn't have a closed-form solution, and would require slower numerical approximations.
> - If a raw GM PDF is put into the denominator, it's likely to be unstable since the denominator can be very small in some regions.
>
> We have already experimented with full GM formulations using adaptive importance sampling, and they are slow and unstable.
> In contrast, using Gaussians is simple and performs well in our experiments.
>
> > 4. Intuition behind this design of guided Gaussian.
>
> In the revised manuscript (Line 183–196), we have added two paragraphs explaining the intuition behind the design.
>
> > 5. Why do you scale the shift $\Delta \mu_n$ by $s_c$?
>
> This is to satisfy bias–variance decomposition, such that when $\tilde{w} = 1$, all the energy in the variance $s_c$ is converted into the bias $s_c \Delta \mu_n$ , making the hyper-parameter $\tilde{w}$ more meaningful. Please refer to the revised manuscript (Line 183–189) for more details.
>
> >  6. How do you obtain $\mu_c$ and $\mu_u$?
>
> In the revised manuscript (Line 174–175 and Appendix A.1), we added that we approximate the conditional and unconditional GM predictions as isotropic Gaussian surrogates by matching the mean and total variance of the GM. This is equivalent to minimizing the KL divergence.
>
> > 7. Orthogonal projection trick
>
> We apply this to $\mu_\text{c} - \mu_\text{u}$, prior to normalization.
>
>  > 8. Please provide the analytical solution of Equation (8).
>
> Your idea is correct. The appendix has discussed conflation of two Gaussians **with powers**. A Gaussian PDF divided by a Gaussian PDF is basically multiplication with power 1 and -1, and the result is still a Gaussian. The code is simply implemented as
> ```
> gm_output = gm_mul_gaussian(gm_cond, gaussian_mul_gaussian(gaussian_guided, gaussian_cond, 1, -1))
> ```
> where all the operations are analytical.
>
> ### **Additional Questions**
>
> > 2. and 5. (Typos)
>
> Thank you for pointing these out! These typos been fixed in the revised manuscript.
>
> > 3. Please explain how you apply DDPM sampling scheme to flow models in details.
>
> Added in the revised manuscript (Appendix A.5).
>
> > 4. Is $\mathcal{L}_\text{spec}$ used for training even if GM-ODE will be used for sampling?
>
> Yes, we use the same model for both ODE and SDE sampling. The spectrum MLP is completely isolated and does not back-propagate to the main model. Therefore, the loss weights do not matter.
>
> > 6. Is there any typo in the definition of ${\mu_s}_k$?
>
> We confirmed that it's correct. This is the same as converting the flow velocity to the score function, i.e., $s_t(x_t) = -\frac{1}{\sigma_t} ( x_t + \alpha_t u)$.
>
> > Is $s$ shared across all spatial locations? What does "the mean of per-pixel GM variances" refer to?
>
> Yes, $s$ is shared. The GM variance refers to the GM’s total variance divided by $D$, which is also dependent on $\{\mu_k\}$. We have added more details in the revised manuscript (Fig. 9).

---

> > ### Comment · Reviewer_A5dP · 2025-04-08
> >
> > Thank you for the detailed explanation. Your responses have addressed my concerns, and I have updated my score accordingly.
> >
> > I have carefully reviewed the entire paper, including the Appendix. In my view, the paper tackles an important problem in flow matching with a novel and well-founded methodology.
> >
> > I believe it has the potential to make a significant impact, and I recommend it for acceptance at ICML.

---

> > > ### Author Response · Authors · 2025-04-09
> > >
> > > Thank you very much for carefully reviewing all the technical details and providing such constructive feedback.

---

### Official Review · Reviewer_gZDF · 2025-03-10

**Overall Recommendation:** 3

**Summary:**

This paper proposes Gaussian mixture flow matching (GMFlow) model, which captures the flow
velocity distribution rather than only predicting its mean based on single-Gaussian assumption. In
addition, the paper utilizes the Gaussian mixture sampling framework to provide probabilistic
guidance via Gaussian mixture Morphing, which alleviate the image over-saturating problem by
avoiding global mean extrapolation. Corresponding SDE and ODE solvers are proposed based on
analytical velocity distributions. Finally, the paper experiments on the 2D checkerboard
distribution and on large-scale class-conditioned ImageNet dataset to show the effectiveness of
the proposed method.

**Claims And Evidence:**

- The novelty of the paper appears to be limited. The GM idea to approximate the reverse
transition kernel of diffusion process is adopted in [1], which is the case of K=2.
- The theoretical analysis of using Gaussian mixture compared to single and bimodal
Gaussian remains to be established. The discretization error reduction could be further
explored with details including the approximation effect of the number of mixing
components K, similar to the Theorem 7 of [2].
- The minimum of the cross-entropy training loss function (Equation (5) and (6)) is not 0
and unknown due to the intractability of the entropy term. The error variance of this loss
can be sub-optimal compared to the modified flow matching loss in [3] that achieves
minimum 0 and reduces the error variance.

**Essential References Not Discussed:**

No, they are adequately discussed.

**Experimental Designs Or Analyses:**

Also see the questions.

**Methods And Evaluation Criteria:**

See the Questions part.

**Other Comments Or Suggestions:**

The theoretical and empirical understanding of choosing suitable number of mixing
components K remains unknown. The theoretical analysis of using Gaussian mixture
compared to single and bimodal Gaussian remains to be established.

**Other Strengths And Weaknesses:**

Strenghts:
- The paper is well organized with clear motives and detailed implementation
considerations.
- The approach to capture the multimodality of velocity distribution by using GM structure
and KL (with the cross-entropy part) training objective is straightforward and analytically
feasible.
- The comparisons and ablation studies are sufficient in small and large scale experiments.

Weaknesses:
- To avoid mode collapse problems of GM models, spectral sampling for image generation
is a detour compared to standard diffusion generation.
- The codes of experiments are not released.

**Questions For Authors:**

- In Fig. 5 and Table 2, could the authors provide some explanations on why the best
Precision drops from K=8 to K=16? It seems to me that larger K should increase the
velocity distribution approximation capability and obtain better results. In addition, why
doesn’t the authors report the results of K=2, which correspond to the case of [1]?
- The effect of larger K is evident in the 2D checkerboard experiment presented by Figure
8(a), while Fig. 5 showed that increasing K has limited positive and even negative effect. Is
this because multimodality is not apparent in the large-scale class-conditioned ImageNet
experiment? If yes, is there any indicator/scheme (heuristic or analytic) for choosing
suitable K in experiments?
- Which “base model” is GMFlow compared to in the “Inference time” paragraph (line
376)? What does “simple” exactly mean in “based on simple arithmetic operations” (line
374), since the calculations in other SDE and ODE solvers are also based on arithmetic
operations that are not complex?
- Is it possible to design another modified flow matching loss similar to [2] that achieves
minimum 0 and lower error variance?
- Is it possible to scale the experiment to large dimensions and preserving spatial
correlations in a more natural way without using spectral sampling?


[1] Hanzhong Guo, Cheng Lu, Fan Bao, Tianyu Pang, Shuicheng Yan, Chao Du and
Chongxuan Li. Gaussian Mixture Solvers for Diffusion Models. In NeurIPS, 2023. arXiv
preprint arXiv:2311.00941.

[2] Tom Huix, Anna Korba, Alain Durmus and Eric Moulines. Theoretical guarantees for
variational inference with fixed-variance mixture of gaussians. In ICML, 2024. arXiv
preprint arXiv:2406.04012.

[3] Gleb Ryzhakov, Svetlana Pavlova, Egor Sevriugov and Ivan Oseledets. Explicit Flow
Matching: On The Theory of Flow Matching Algorithms with Applications. arXiv preprint
arXiv:2402.03232.

**Relation To Broader Scientific Literature:**

The paper applies the GM framework to GM morphing, which is an appealing direction in
CFG to mitigate OOD problems.

**Theoretical Claims:**

I've read the proof for most theorems but not checked them carefully.

---

> ### Author Rebuttal · Authors · 2025-04-01
>
> Thank you for your thoughtful review. We have uploaded a **revised manuscript** and essential **code** in this anonymous link (full code will be released upon publication):
>
> https://anonymous.4open.science/r/anonymous_gmflow-63FE
> backup: https://limewire.com/d/CgAn9#jkBxDmC3qh
>
> > The GM idea is adopted in [1]
>
> Please note that [1] is essentially a moment matching method, which only converts its moment predictions into a bimodal GM during inference. This is fundamentally different from our GMFlow formulation, which directly learns GM parameters. In comparison, [1] employs three L2 losses for three moments, whereas we use a single loss for all Gaussian components.
>
> Moreover, our formulation can generalize to more GM components, and we further propose few-step ODE sampling with analytical substeps and probabilistic guidance.
>
> > Theoretical analysis of using GM
>
> Gaussian mixtures are well known to yield better approximation with more components [2, 4]. In diffusion SDE sampling, it is also widely recognized that improving the accuracy of the reverse transition distribution can greatly improve few-step sampling. Taken together, these observations already support the claim that using more Gaussian components should lead to reduced SDE sampling errors.
>
> The practical performance ultimately depends on how the network learns the GM parameters, a process that is difficult to analyze theoretically. Therefore, we believe that our empirical validation through experiments is more important here.
> - [2] Huix et al. Theoretical guarantees for variational inference with fixed-variance mixture of gaussians.
> - [4] Bishop. Mixture Density Networks
>
> > Minimum of the cross-entropy training loss
>
> Cross-entropy equals the sum of the data entropy and the KL divergence. Although the data entropy is intractable, it is solely determined by the data and independent of the model, thus it can be ignored in the loss function. Note that Eq. (6) omits all irrelevant constant terms.
>
> Our goal is to minimize the KL divergence between the predicted GM and the ground truth distribution, which has a minimum of 0 when the GM perfectly fits the ground truth.
>
> > Flow matching loss in [3] achieves minimum 0 and reduces the error variance
>
> In general, the flow matching loss does not have a tractable form. While [3] analyzes special cases with analytic velocity fields, our work focuses on practical applications.
>
> Theoretically, the "error variance" of stochastic flow matching loss equals the sum of denoising distribution variance and the squared error of the velocity. The former is determined by the data, whereas the latter can be reduced to near 0 with enough capacity.
>
> Different from standard flow matching, our goal is not just to learn accurate local velocity, but also to capture the underlying denoising distribution so that a global velocity field can be analytically derived for multi-substep GM-ODE sampling.
>
> > Spectral sampling for image generation is a detour
>
> We want to clarify that spectral sampling is just an **optional extension** to standard diffusion sampling with **minimal overhead**.
> - **Optional**: It's not used in our ODE solvers. In our SDE solvers, it's an optional trick to improve FID, whereas Precision is not affected, as shown in our ablation studies (Table 4).
> - **Extension**: When setting $K=1$ and using a uniform spectrum, spectral sampling is equivalent to standard diffusion sampling. Therefore, **we consider it a more general approach instead of a detour**.
> - **Minimal overhead**: The total overhead of GMFlow is less then 2% in inference time.
>
> > Choosing suitable K
>
> With pixel-wise factorization, multimodality is indeed less apparent. From a practice point of view, we choose the best $K$ based on the evaluation metrics of interest. We also found that early-step training NLL can serve as a potential indicator.
>
> > Why the best Precision drops from K=8 to K=16
>
> We believe this is due to numerical errors in spectral sampling, which grow larger with increasing $K$. We have tested the case without spectral sampling, where both $K=8$ and $K=16$ yield the same Precision of 0.946, with NFE=8.
>
> > Results of K=2
>
> We have added an experiment using $K=2$ on the CIFAR-10 dataset (training on ImageNet is beyond the time frame of rebuttal), and our few-step results are much better than GMS. Please refer to our response to **Reviewer A5dP**.
>
> > Inference time
>
> Sorry for the confusion. We have rewritten this paragraph for improved clarity: GMFlow adds only 0.005 sec of overhead per step (batch size 125, A100 GPU) compared to its flow-matching counterpart, which is minimal compared to the total inference time of 0.39 sec per step---most of which is spent on DiT.
>
> > Scale the experiment to large dimensions and preserving spatial correlations
>
> This will be one of our future directions. We have recently conducted preliminary experiments using patch-wise factorization instead of pixel-wise factorization, which show promising results.

---

### Official Review · Reviewer_qy1B · 2025-03-12

**Overall Recommendation:** 4

**Summary:**

The authors present a new formulation of diffusion models, termed as Gaussian mixture flow matching (GMFlow). Unlike existing diffusion models, GMFlow models the PDF of velocity by predicting the parameters of a Gaussian mixture (GM) distribution. Based on this formulation, GMFlow can generate high-quality images with fewer sampling steps. To avoid over-saturation artifacts, the authors propose a probabilistic guidance via GM morphing. In addition, the authors designs specific SDE/ODE solvers for GMFlow. They validate the effectiveness of the proposed method on both 2D toy datasets and ImageNet.

## update after rebuttal
I appreciate their thorough response and the rebuttal has addressed my concerns. I raise my score accordingly.

**Claims And Evidence:**

Strengths
+ The claims regarding the limitations of diffusion and flow models are correct and wildly recognized in the field of generative models.
+ The proposed method is reasonable  and easy to understand. It enhances the representation capacity of flow models and enables high-quality generation with fewer steps.

Weaknesses

None

**Essential References Not Discussed:**

To my knowledge, there is a work that should be discussed. The authors should include a discussion of GIVT [1] in the related work.

[1] Tschannen, Michael, Cian Eastwood, and Fabian Mentzer. "Givt: Generative infinite-vocabulary transformers." European Conference on Computer Vision. Cham: Springer Nature Switzerland, 2024.

**Experimental Designs Or Analyses:**

Strengths
+ The superior results on both 2D toy datasets and ImageNet demonstrate the effectiveness of GMFlow. This paper also provides exhaustive ablation studies to assess each key component in Table 4.
+ The qualitative results across different sampling steps (Figure 2,6, and 8) make the efficiency of GMFlow more apparent and intuitive.

Weaknesses
- The evaluations on visual generation are insufficient. It is better to evaluate the proposed method across different model sizes and image sizes. For model architecture, DiT-L/2 is an option. And, the evaluating on ImageNet $512 \times 512$ would help demonstrate the generality of the proposed method.
- Altough Inception Score (IS) is mentioned in Sec. 4.2 (Evaluation protocal), no results are provided. It would be beneficial to include the IS metric to comprehensively assess the proposed method.

**Methods And Evaluation Criteria:**

Strengths
+ The proposed method is both intuitive and reasonable. Modeling the PDF of velocity enhances the representation capacity of flow models, leading to more effective modeling of complex distribution. Probabilistic guidance and SDE/DOE solvers are specifically designed to address the challenges of expensive sampling steps and over-saturation artifacts.
+ The evaluation benchmark is reasonably appropriate for assessing the effectiveness of the proposed method.

Weaknesses
- The proposed method relies on a pre-defined number of mixtures, which can limit the types of distributions. In addition, using an excessive number of mixtures leads to high costs and requires a large model for effective representation.

**Other Comments Or Suggestions:**

None

**Other Strengths And Weaknesses:**

None

**Questions For Authors:**

I acknowledge the novelty of the proposed method, but I also have concerns regarding the pre-defined number of mixtures and the insufficient evaluations. Thus, I am inclined to rate this paper as weak accept.

**Relation To Broader Scientific Literature:**

To my knowledge, the proposed method in this paper is new.

**Theoretical Claims:**

I have check the correctness of any proofs for theoretical claims. And I do not find any major errors.
typos: In Equation (6), $x$ and $\mu_k$ are related to $x_t$, and since they are derived from the density of $\mathcal{N}(\mu; \mu_k, s^2)$, it would be better written as $\| \mu(x_0, x_k) - \mu_k(x_k)\|^2$.

---

> ### Author Rebuttal · Authors · 2025-04-01
>
> Thank you for your thoughtful review. We have uploaded a **revised manuscript** and essential **code** in this anonymous link (full code will be released upon publication):
>
> https://anonymous.4open.science/r/anonymous_gmflow-63FE
> backup: https://limewire.com/d/CgAn9#jkBxDmC3qh
>
> > Pre-defined number of mixtures can limit the types of distributions. Using an excessive number of mixtures leads to high costs and requires a large model
>
> With pixel-wise factorization, a small number of Gaussians is enough to capture the distribution of a single latent pixel. The experiments also reveal that image generation metrics generally saturate at $K=8$.
>
> On the other hand, GMFlow only expands the output channel of the final layer to accommodate the mixture components. With $K=8$, currently, the final layer has an output channel size of  $(4 + 1) \times K = 40$, where 4 is the latent channel size. So the computational cost of increasing K is still minimal when compared to that of the remaining layers in DiT, especially considering the hidden dimensions of DiT are usually much larger.
>
> > typos
>
> Thank you for pointing this out. We have fixed the typos in the revised manuscript.
>
> > Evaluating the proposed method across different model sizes and image sizes
>
> We have added an experiment on unconditional CIFAR-10 image generation (please refer to our response to **Reviewer A5dP**), which employs different architecture (U-Net), model size (53M), and image resolution (32x32) from the DiT used in the paper.
> Unfortunately, training 512-res GMFlow and baseline methods are too computationally expensive and time-consuming, and is beyond the time frame of the rebuttal.
>
> > IS metric
>
> Following previous work [3, 4], we employ Precision as the quality measurement instead of IS. This is because IS metric does not capture the image quality under a large guidance scale---in particular, IS is not sensitive to over-saturation. In fact, both our GMFlow model and flow matching baseline reach IS > 500 under the highest guidance scale evaluated, despite that the baseline has clearly worse visual quality.
>
> - [3] Sadat et al. Eliminating Oversaturation and Artifacts of High Guidance Scales in Diffusion Models
> - [4] Kynkäänniemi et al. Applying Guidance in a Limited Interval Improves Sample and Distribution Quality in Diffusion Models
>
> > Discussion of GIVT
>
> Thank you for suggesting this interesting paper. In the revised manuscript, we have added a paragraph discussing GIVT and other generative models using GMs.

---

### Official Review · Reviewer_wKf5 · 2025-03-17

**Overall Recommendation:** 4

**Summary:**

This paper proposes a Gaussian mixture (GM) flow matching (FM) model. The traditional FM model uses a Gaussian modeling velocity field, while the proposed GMFlow method in this paper uses a Gaussian mixture modeling velocity field. The author shows that GMFLow can produce better results with fewer steps. The author also proposes an SDE/ODE sampling algorithm suitable for GMFlow, as well as a probabilistic CFG algorithm, which can alleviate the over-saturation problem caused by traditional CFG.

## update after rebuttal
I thank the authors for their response and I will maintain my score as Accept.

**Claims And Evidence:**

The paper claims that GMFlow has a lower number of sampling steps and that probabilistic guidance can alleviate oversaturation, which has been supported by experiments.

**Essential References Not Discussed:**

There is relatively little discussion of related work in this paper, and it is suggested to increase the discussion on the combination of GM with other generative models such as VAE.

**Experimental Designs Or Analyses:**

The experimental design is mostly reasonable, especially the 2D experiment clearly shows the advantage of GMFlow in few steps.
The experiment in Table 2 is not perfect. The results show that increasing K has benefits, but the boundary is not seen.

**Methods And Evaluation Criteria:**

Yes.

**Other Comments Or Suggestions:**

- In Theorem 3.1, w.r.t. with
- In Line 70, $x_T \approx \epsilon$ does not take into account VE diffusion

**Other Strengths And Weaknesses:**

Strengths
+ The method in this paper is novel and promising to me
+ The method proposed in this paper has many technical contributions. In addition to combining GM and Flow matching, it also proposes probabilities guidance and a new SDE/ODE sampler

Weaknesses
- Table 3 is the main experimental result, and FID is not used as a reference indicator

**Questions For Authors:**

- Will the results be better if $sI$ is replaced by a diagonal matrix?
- Does pixel-wise factorization mean that each latent pixel is treated as a Gaussian component? How does this relate to K?
- Does pixel-wise factorization limit the method to a fixed resolution setting?

**Relation To Broader Scientific Literature:**

No.

**Theoretical Claims:**

I have currently checked the correctness of  Theorem 3.1 in the main paper.

---

> ### Author Rebuttal · Authors · 2025-04-01
>
> Thank you for your thoughtful review. We have uploaded a **revised manuscript** and essential **code** in this anonymous link (full code will be released upon publication):
>
> https://anonymous.4open.science/r/anonymous_gmflow-63FE
> backup: https://limewire.com/d/CgAn9#jkBxDmC3qh
>
> > Boundary of experiment in Table 2
>
> Unfortunately due to limited computing resources, we were not able to conduct experiments for K values beyond 16. That being said, as shown in Fig. 5 (or Fig. 7 in the revised manuscript). we have shown that the FID and Precision metrics generally saturate when $K\ge 8$. The NLL values in Table 2 do not directly reflect image generation quality, and they are more relevant to some downstream applications (e.g., score distillation-like test-time optimization), which is not the main focus of this work. We are happy to add more discussions in the final draft.
>
> >  Discussion on the combination of GM with other generative models such as VAE
>
> Thank you for the suggestion. In the revised manuscript, we have added a paragraph discussing GM GANs and autoregressive Transformers. In the context of modern generative AI, however, VAEs alone are often regarded not as generative models but rather as representation compressors. Some GM VAE papers are focused on clustering instead of generation [1, 2]. We are happy to add more related works in the final draft.
>
> - [1] Dilokthanakul et al. Deep Unsupervised Clustering with Gaussian Mixture Variational Autoencoders.
> - [2] Jiang et al. Variational Deep Embedding: AnUnsupervised and Generative Approach to Clustering
>
> > FID in Table 3
>
> In Table 3, we aim to evaluate the saturation of different methods at their best precision, which typically requires large CFG values that are beyond reasonable ranges for FID comparison. The main comparisons on both FID and Precision are presented in Fig. 4 (or Fig. 6 in the revised manuscript).
>
> > In Line 70, $x_T \approx \epsilon$ does not take into account VE diffusion
>
> Thank you for pointing this out. The manuscript states "A typical diffusion model", which does not cover all cases. In practice, mainstream diffusion models are trained with VP schedules, and they can be rescaled into VE diffusions during sampling (which is how the popular EDM Euler solver is implemented).
>
> > Will the results be better if $sI$ is replaced by a diagonal matrix?
>
> We have tried predicting pixel-wise variances instead of the global $s$ in image generation. The cons outweigh the pros: it makes training less stable since it's more likely to have small variances in the denominator of the loss function, and the benefits for NLL can be equally achieved by increasing $K$.
>
> > Does pixel-wise factorization mean that each latent pixel is treated as a Gaussian component? How does this relate to K?
>
> Yes. Each latent pixel (4 channels) is a 4-D GM of $K$ components. Please refer to the network architecture in Fig. 3 (or Fig. 4 in the revised manuscript).
>
> > Does pixel-wise factorization limit the method to a fixed resolution setting?
>
> We don’t think so. We think the opposite might be true: without pixel-wise factorization, the entire latent grid would be treated as a high-dimensional GM, which would complicate diverse resolution generation due to varying dimensions of GM. We think pixel-wise factorization makes diverse resolution generation easier since the per-pixel dimensions are fixed.

---

### Decision · Program_Chairs · 2025-05-01

**Decision:**

Accept (poster)

**Comment:**

All reviewers have carefully assessed the paper and provided detailed comments on its contributions. The authors have also addressed all the concerns regarding technical details and references to prior work. The reviewers have acknowledged the rebuttals and integrated them in their reviews. The overall recommendation from all reviewers is to accept the paper. All acknowledge that the paper is a clear step forward with several innovations (eg, a faster generation process and a solution to the over-saturation problem caused by traditional CFG) to the training of generative models for flow matching.